# Curing Sustainability Assessment in Concrete Pavements: A 20-Year Simulation-Based Analysis in Urban Road Contexts

Julián Pulecio-Díaz 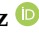

Faculty of Engineering, Universidad Cooperativa de Colombia, Medellin Edificio I, Ibague 730006, Colombia; julian.puleciod@campusucc.edu.co

**Abstract:** In urban areas with warm climates, a lack of proper curing during concrete pavement construction can significantly reduce service life, increase maintenance needs, and compromise sustainability goals. Despite its relevance, the comprehensive impact of curing has been poorly quantified from a multidimensional perspective. This study aims to evaluate the effect of applying a liquid curing compound on the sustainability of concrete slab pavements over a 20-year horizon using a simulation-based approach. Two scenarios, cured and uncured, were modeled with HIPERPAV®, incorporating site-specific climatic, structural, and material parameters. Based on projected maintenance cycles, nine sustainability indicators were calculated and grouped into environmental ($CO_2$ emissions, energy, water, and waste), social (accidents, travel time, satisfaction, and jobs), and economic (life-cycle maintenance cost) dimensions. Statistical tests (ANOVA, Welch ANOVA, and Kruskal–Wallis) were applied to assess significance. Results showed that curing reduced $CO_2$ emissions (−13.7%), energy consumption (−12.5%), and waste (−20.7%), while improving accident rates (−40.3%), user satisfaction (+17.8%), and maintenance cost savings (−9.5%). The findings support curing as a cost-effective and sustainability-enhancing strategy for urban pavement design and management.

**Keywords:** sustainability of road infrastructure; long-term sustainability assessment; concrete slab pavements; concrete curing methods



## 1. Introduction

The need to develop sustainable road infrastructure has gained relevance in the face of global challenges related to climate change, resource efficiency, and urban quality of life [1]. This approach led to establishing concrete slab pavements as a durable and resistant alternative for medium and high-traffic urban areas, thanks to their low maintenance and good structural performance [2,3].

However, the long-term performance of these pavements depends largely on the curing conditions during the early stages of construction. Proper concrete curing allows complete cement hydration, improving its mechanical strength, reducing crack formation, and prolonging the service life of the pavement [4–7]. Research shows that no or inadequate curing can significantly decrease durability and increase the frequency of maintenance interventions [8,9].

Recent studies also highlight that curing efficiency directly influences wear resistance, concrete microstructure, and protection against freezing–thawing cycles [10–12]. Appropriate curing method selection is crucial to reduce chloride penetration and improve pavement life, especially in harsh urban environments [13]. Furthermore, research has demonstrated

that innovative methods such as carbonation curing improve concrete compactness and resistance to external agents [14].

From an environmental perspective, curing also has significant implications [15]. Life cycle assessment studies show that poorly cured pavements increase total carbon footprint due to more frequent repairs and higher utilization of materials [16–18]. On the other hand, efficient curing reduces energy consumption and waste generated by extending maintenance intervals [19–21]. Furthermore, improved durability has been documented to reduce the frequency of rehabilitations, minimizing indirect $CO_2$ emissions [22].

Along this line, Hou et al. (2024) [14] demonstrated that an efficient curing strategy, combined with a sustainable mixture design (e.g., using recycled fibers and alternative sands), reduces the overall environmental impact by up to 35% compared to conventional solutions. These findings highlight the need to integrate curing as a critical variable in the sustainability analysis of urban pavements.

At the social level, it has been shown that an improved pavement condition is associated with lower accident rates, greater user comfort, and reduced travel times, key factors in user perception of the quality of the urban environment [23–26]. Furthermore, curing treatment implementation may require more technical labor, favoring job creation in the construction and maintenance phases [27].

Authors such as Ghosh and Banerjee (2025) [28] have explored stabilizing materials and additives that, combined with appropriate curing processes, significantly improve road safety and the structural response of urban soils, even under expansive conditions. This evidence reinforces the idea that curing impacts the service life and the functional safety of pavements in real-life scenarios.

In economic terms, several studies conclude that, although curing represents an additional initial cost, it translates into long-term net savings due to reductions in the frequency and severity of corrective interventions [29–32]. Likewise, the life-cycle cost approach has proven to be a valuable tool for justifying technical decisions prioritizing durability and operational efficiency [33,34].

In contexts where the search to transition toward low-carbon concretes is a priority, such as the one developed by Singh et al. (2024) [35], by using coal bottom ash, the authors found that curing is crucial for ensuring material functionality and avoiding operating cost overruns due to premature failure. These findings expand the scope of curing as a key preventive mechanism in emerging technologies.

Despite the advances identified in the literature, a limited integration of the effects of curing from a comparative and comprehensive perspective that simultaneously considers the economic, environmental, and social dimensions persists. Therefore, this article aims to evaluate the impact of curing in slab pavements on its sustainability through a quantitative comparison between scenarios with and without curing over a 20-year analysis horizon, applied to the urban context of Comuna 8 (sector 8) in Ibagué, Colombia.

This methodology combines structural simulation with multi-criteria impact analysis in three key dimensions, allowing for a robust long-term sustainability assessment in Latin American urban contexts.

## 2. Methodology

The methodological approach of this study combines computational simulation, statistical analysis, and sustainability criteria to evaluate the impact of curing on the service life of concrete slab pavements.

The methodology is structured in six interconnected phases (Figure 1). First, pavement service life is estimated using HIPERPAV® simulations [36,37], considering the early

behavior of the concrete and its long-term structural performance. Then, sustainability variables and indicators are selected based on the existing literature and project criteria.

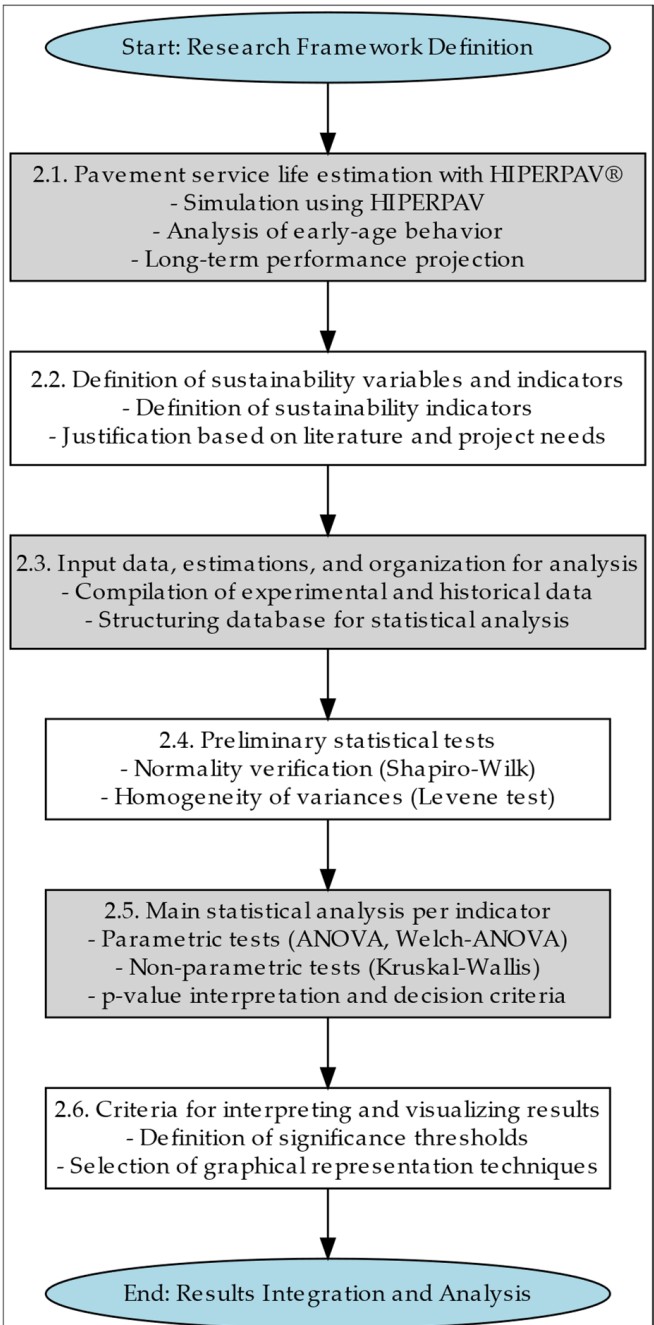

**Figure 1.** Methodological framework to assess the curing impact on concrete slab pavements.

Subsequently, data are collected and organized, including experimental information and historical records. Comuna 8, located in Ibagué, Colombia, was selected as the representative urban area for the methodological approach to be applied, due to the availability of climatic and structural data, and as a relevant subject for testing the long-term impact of construction practices in Latin American cities. This area has a warm mid-altitude climate, intermediate traffic loads, and recurrent pavement maintenance needs. Although the findings are based on this particular context, the methodology developed, including the pavement performance simulation and the multi-indicator sustainability framework, is fully adaptable to other urban settings with similar characteristics.

Statistical conditions are assessed using normality (Shapiro–Wilk) and homogeneity of variance (Levene) tests [38,39]. Based on these results, parametric (ANOVA and Welch-ANOVA) or nonparametric (Kruskal–Wallis) tests are applied, establishing decision criteria based on the *p*-value [40,41]. Finally, interpretation and visualization strategies are defined, allowing for the integration of results in terms of sustainability.

### 2.1. Pavement Service Life Estimation with HIPERPAV®

The service life of concrete slab pavements in scenarios with and without curing was estimated through computer simulation using the HIPERPAV III® – Version 3.2 software (HIgh PERformance Concrete PAVing) developed by the Federal Highway Administration (FHWA). This tool was designed to predict early concrete behavior and its impact on long-term pavement performance, integrating thermal, mechanical, structural, and environmental aspects.

The simulation process included two fundamental phases. First, the early behavior of the concrete was modeled, with emphasis on crack formation due to thermal and drying shrinkage, as well as the strength evolution during the first hours and days of placement. This analysis allows the anticipation of early cracking problems that affect the future durability of the pavement. Second, a projection of structural behavior was made throughout the functional life cycle, considering the cumulative equivalent single axle loads (ESALs) based on the expected loading conditions over an analysis horizon of 20 years. This combined simulation allowed the estimation of the effective pavement performance periods before requiring maintenance interventions.

The models were fed with representative project data, including the following:

- Climatic conditions: Ambient temperature, relative humidity, solar radiation, wind, and initial concrete temperature.
- Pavement structural design: Slab thickness, base type, subgrade reaction modulus, and joint distribution.
- Curing characteristics: Presence or absence of surface liquid application.
- Concrete properties: Obtained from experimental tests of compressive resistance, modulus of elasticity, hydration heat, and coefficient of thermal expansion.

Based on this information, the simulator estimated the frequency of interventions required in each scenario, allowing several maintenance cycles to be associated with a 20-year horizon. Following a life-cycle approach, this estimate was key to calculating the cumulative impacts on each defined sustainability indicator.

Figure 2 illustrates the methodological framework implemented to simulate the service life of the pavement. The process begins with the integration of input data, including climatic conditions (e.g., temperature, humidity, wind, and radiation), pavement design parameters (e.g., geometry, slab support, joint design, traffic pressure), and concrete mix properties (e.g., PCC composition, cement content, water–cement ratio, and type of curing). These inputs feed the HIPERPAV® simulation environment. The first phase focuses on early-age concrete behavior, analyzing the risk of cracking due to thermal and drying shrinkage, as well as strength development in the initial hours and days. The second phase projects the long-term structural performance of the pavement over a 20-year horizon, considering traffic load accumulation and estimating maintenance and intervention cycles. This two-phase simulation approach provides a technically robust and context-sensitive representation of pavement deterioration, aligned with the sustainability indicators evaluated in subsequent sections.

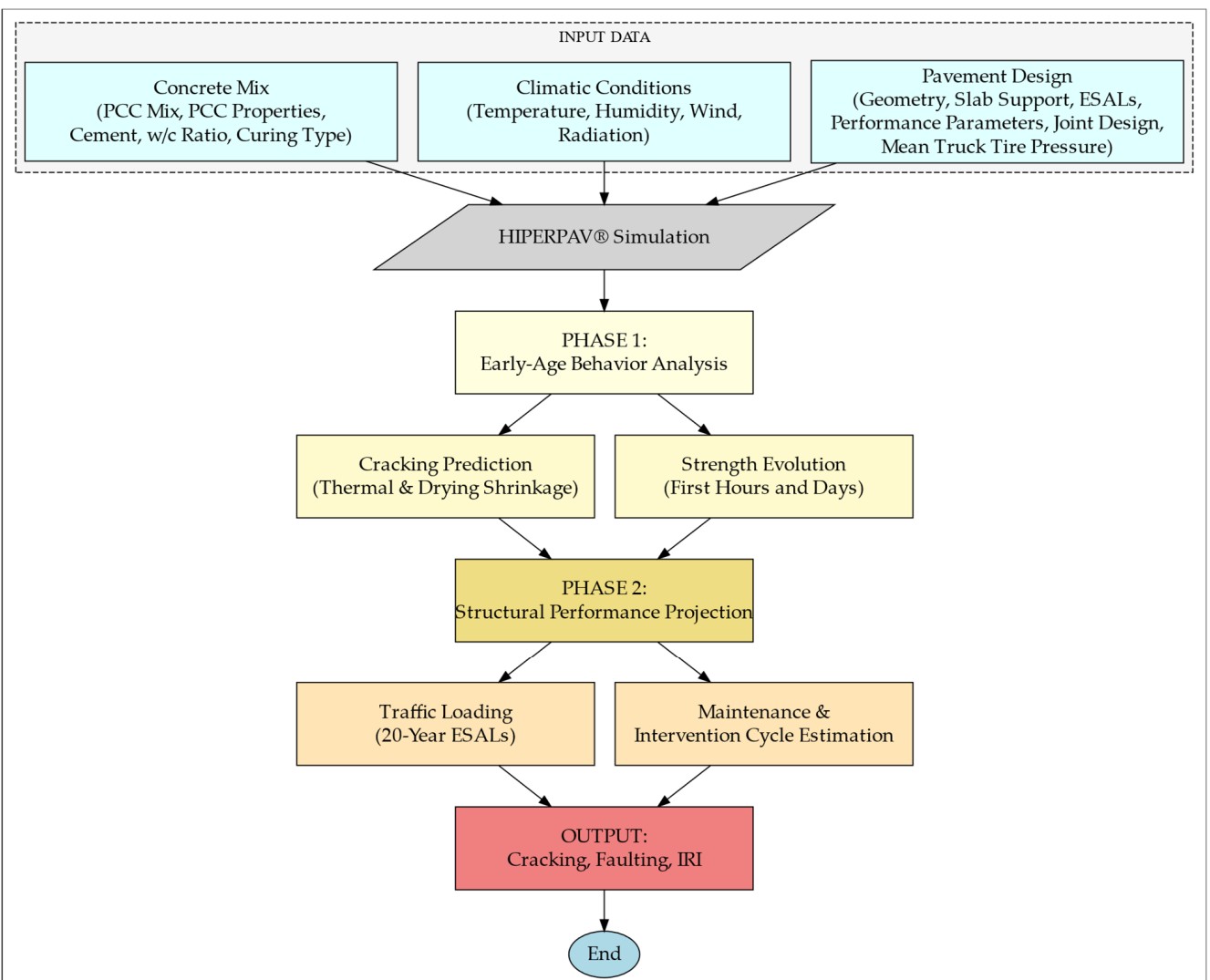

**Figure 2.** Flowchart of the pavement service-life estimation process through a simulation in HIPERPAV®.

## 2.2. Definition of Sustainability Variables and Indicators

To comparatively assess the sustainability of concrete slab pavements under cured and uncured conditions, a set of nine indicators was defined and classified according to the three fundamental dimensions of sustainable development: (i) environmental, (ii) social, and (iii) economic. This classification aligns with the life cycle analysis approach and enables the interpretation of the cumulative effects of the design and the maintenance strategy on overall pavement performance.

The indicators were selected based on the following:

- Their relevance in road infrastructure sustainability studies;
- Their ability to be quantified in homogeneous units per kilometer of road;
- Their sensitivity to the number of interventions required during the analyzed time horizon;
- The possibility of obtaining them through reliable technical sources and input data, consistent with the HIPERPAV® simulation.

Each indicator represents a direct manifestation of pavement performance in a key sustainability dimension and was selected based on criteria of relevance, sensitivity to maintenance conditions, and compatibility with life-cycle assessment frameworks such as

ISO 14040 and Envision™ – Version 2.9.5. Indicators were quantified using standardized procedures and expressed in comparable units per kilometer. Environmental values were derived from emission and resource consumption factors reported in technical literature. Social metrics were based on urban mobility data, perception surveys, and labor estimates associated with maintenance activities. Economic costs were calculated using unit values applied to the maintenance cycles projected by HIPERPAV®. The assumptions, sources, and parameter values are detailed in the corresponding tables within the results section.

The purpose and unit of analysis for each indicator are described below.

### 2.2.1. Environmental Indicators

Environmental impacts are related to the use of resources and the generation of externalities associated with pavement construction and maintenance. The selected indicators allow quantifying the effects on the physical environment in terms of input consumption and emissions:

- Carbon dioxide emissions (kg $CO_2$/km): This indicator quantifies the carbon footprint generated by using cement, materials, transport, machinery, and energy consumption during construction, maintenance, and rehabilitation activities. It was estimated using standardized emission factors by activity.
- Energy consumption (MJ/km): This indicator represents the direct and indirect energy involved in the life cycle of the pavement. It includes the energy embodied in the materials and the machinery used during interventions.
- Water consumption ($m^3$/km): This indicator measures the amount of water used during concrete manufacturing, the curing process, and other auxiliary construction activities. This resource is particularly critical in urban contexts with water stress.
- Construction waste (kg/km): This indicator refers to materials discarded due to the partial or total demolition of pavement sections and maintenance activities. It was calculated based on the volume removed and the density of the materials.

### 2.2.2. Social Indicators

The social effects of pavement are related to user experience, road safety, and the impact on urban dynamics. The indicators used seek to reflect how the condition of the infrastructure influences life quality and citizen perception:

- Accident reduction (%): This indicator reflects the relative change in the accident rate attributable to improved pavement surface condition, reduced crack formation, and greater surface evenness. This estimate was based on the technical literature, considering the correlation between pavement quality and accident rates.
- Travel time (minutes): This indicator assesses the impact of pavement on urban mobility, considering both superficial conditions and maintenance frequency, which involves closures or detours. This indicator was expressed as cumulative travel time per kilometer.
- Citizen satisfaction (%): This indicator represents user perception of the service quality of road infrastructure. Although it is a subjective variable, it can be estimated indirectly through technical proxies, such as the functional condition of the pavement and the regularity of interventions.
- Job creation (persons/year): This indicator estimates the number of jobs required per year for construction and maintenance activities. This indicator reflects the social impact of pavement as a generator of local economic activity.

### 2.2.3. Economic Indicator

The economic indicators focused on projected life cycle costs associated exclusively with maintenance interventions, as the objective of the study is to compare the effect of curing in the initial post-construction phase.

Cumulative maintenance cost (COP/km): This indicator is calculated from the estimated number of interventions over 20 years and unit costs by activity type (minor repairs, major repairs, or rehabilitation). This indicator is expressed in Colombian pesos (COP) per kilometer and represents the total expenditure required to maintain pavement functionality in acceptable conditions.

Each indicator was calculated for both scenarios (i.e., with and without curing), considering the number of interventions estimated using the HIPERPAV® simulator. The quantification of cumulative impacts established a solid and technically consistent comparative basis for subsequent statistical analysis, ensuring a comprehensive sustainability approach aligned with frameworks such as life cycle assessment (LCA) [42] and multi-criteria urban infrastructure assessment.

### 2.3. Input Data, Estimations, and Organization for Analysis

Data collection and structuring were carried out systematically to ensure the traceability and validity of the inputs used in the simulation and comparative analysis. The process consisted of three stages: (i) obtaining technical and experimental data to feed the HIPERPAV® model, (ii) estimating the cumulative impacts by indicator, and (iii) organizing the data into comparative statistical matrices.

### 2.3.1. Input Data for HIPERPAV® Simulation

Pavement behavior simulations were performed using the HIPERPAV® software, which requires detailed parameters of the climatic environment, the structural design of the pavement, and the concrete mixture. Data were obtained through a combination of the following:

- Historical meteorological sources (local stations and climatological databases) were used to determine daily ambient temperature, relative humidity, wind speed, and solar radiation. These parameters enable modeling the heat and moisture exchange that influence concrete hydration and shrinkage.
- Structural design data, including slab thickness, granular or stabilized base type, subgrade reaction modulus (k), slab dimensions, joint spacing, and traffic load type. This information came from typical technical specifications for medium-traffic urban roads supplemented by road design manuals. Concrete properties, determined in the laboratory using standardized tests (ASTM C39 [43] and C469 [44], among others), included compressive strength, modulus of elasticity, heat of hydration, drying shrinkage, coefficient of thermal expansion, and water/cement ratio (w/c ratio). These properties were used to represent the mechanical and hygrothermal behavior of the material accurately.
- Curing conditions: Two contrasting scenarios were modeled. In the first, the omission of surface curing was simulated; in the second, applying a liquid curing compound in a single layer was considered.

Based on these data, HIPERPAV® estimated the early behavior of the concrete (temperature and shrinkage cracking, development of initial strengths) and its impact on projected durability. Based on the performance and fatigue analysis, the model estimated the number of intervention cycles expected over a 20-year horizon, differentiating between the two scenarios.

### 2.3.2. Estimation of Cumulative Impacts

Once the projected intervention cycles were obtained, the cumulative impacts on the nine defined sustainability indicators were estimated. Unit impact values per cycle (per kilometer of pavement) were determined using secondary sources, technical databases, and conversion factors from the scientific literature, technical reports, and previous LCA studies on road infrastructure.

For each indicator, the following calculation logic was applied (see Equation (1)):

$$\text{Cumulated impact (20 years)} = \text{Impact per cycle} \times \text{Number of estimated cycles in HIPERPAV}^{\circledR} \tag{1}$$

This allowed the standardization of the results for both scenarios (with and without curing), expressing them in comparable units per kilometer in the following domains:

Environmental: $CO_2$ emissions (kg/km), energy consumption (MJ/km), water consumption ($m^3$/km), and waste generation (kg/km).

Social: Accidents avoided (%), cumulated travel time (minutes), citizen satisfaction (%), and job creation (persons/year).

Economic: Cumulative maintenance cost (COP/km), updated using construction price indices and projected growth rates.

### 2.3.3. Data Organization and Preparation for Statistical Analysis

The data were organized into a two-way matrix structure, with the nine indicators as dependent variables and the two scenarios (with and without curing) as independent groups, to perform a robust and transparent statistical analysis.

The observations were structured into grouped lists for each indicator, calculating the impacts under the same functional unit and time horizon conditions. Then, the data were reviewed for integrity, numerical consistency, and coherence with the assumptions of the HIPERPAV® model.

Variable normalization was not applied, as the statistical analysis was performed independently by variable (univariate), allowing the actual units of each indicator to be preserved, facilitating technical interpretation and the practical utility of the results.

This organization allowed the subsequent application of appropriate statistical tests (ANOVA, Welch ANOVA, or Kruskal–Wallis), selected based on the distribution and variance of each data set, as detailed in the following section.

### *2.4. Preliminary Statistical Tests*

Prior to applying inferential tests, a rigorous evaluation of the statistical conditions for each variable was conducted. This was carried out to select the most appropriate tests to compare the indicators between the two analysis scenarios: cured and uncured pavements. This stage was essential to ensure the validity and robustness of the results obtained in the univariate analysis.

The evaluation focused on the two fundamental assumptions that govern the application of parametric tests, such as an ANOVA: data normality and homogeneity of variances between groups.

### 2.4.1. Shapiro–Wilk

The normality distribution of each variable was verified using the Shapiro–Wilk test, which is widely recognized for its sensitivity in small and moderate samples. This test proves the null hypothesis (H0), i.e., that the data show a normal distribution.

To ensure the robustness of the analysis, the following occurred:

- It was applied independently for each indicator and group (with and without curing).
- A significance level of $\alpha = 0.05$ was adopted.
- A $p$-value $< 0.05$ indicated a significant deviation from normality, suggesting that the data do not follow a normal distribution.

Figure 3 illustrates the decision-making process for the Shapiro–Wilk test. A data sample is initially collected, and a normality test is performed. If the resulting $p$-value is higher than or equal to 0.05, the null hypothesis (H0) is not rejected, implying that the data can be considered normally distributed. However, if $p < 0.05$, H0 is rejected, concluding that the data distribution differs significantly from a normal distribution.

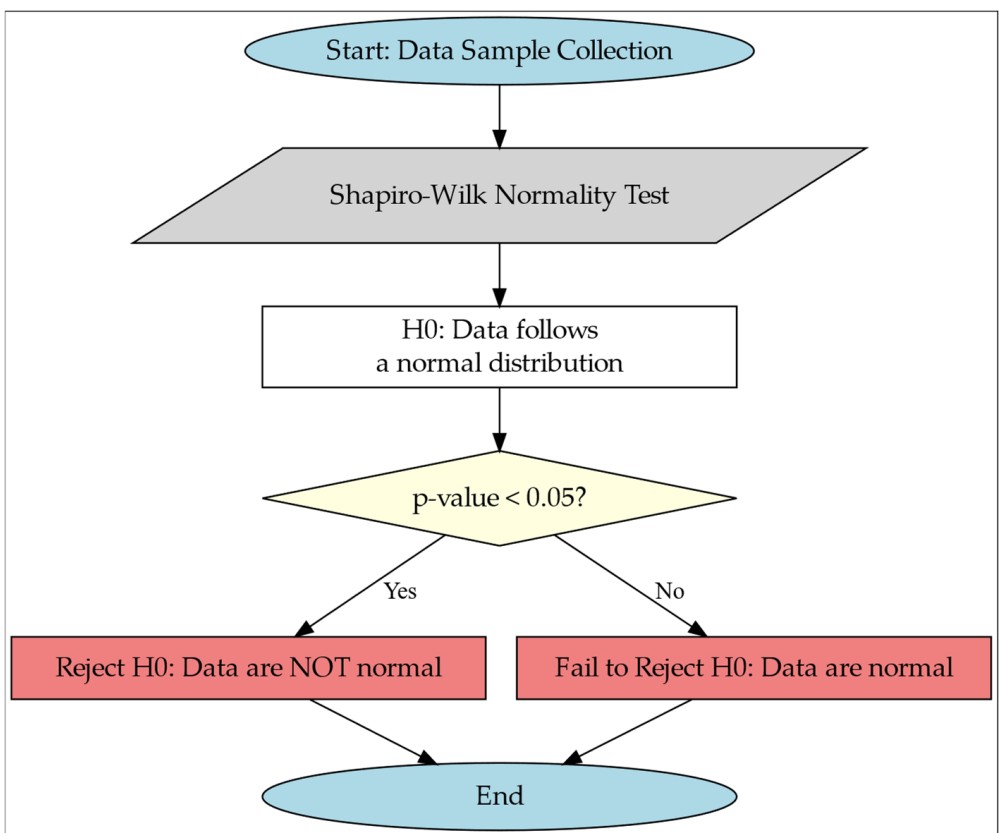

**Figure 3.** Flowchart of the procedure applied for the Shapiro–Wilk normality test.

### 2.4.2. Levene

For indicators that met the normality criterion, Levene's test was applied to verify the equality of variances between both groups. This test is less sensitive to deviations from normality and is considered appropriate for comparing dispersion between groups of unequal size:

- A result of $p \geq 0.05$ indicates that the variances are homogeneous, allowing the application of the traditional ANOVA.
- If the variances are significantly different ($p < 0.05$), the traditional ANOVA is discarded in favor of a robust alternative such as the Welch ANOVA.

Figure 4 illustrates Levene's test procedure using a flowchart. It begins with data collection and the test application under the null hypothesis (H0) that the variances between the groups are homogeneous. The $p$-value is then evaluated; if it is higher than or equal to 0.05, H0 is not rejected, concluding that the variances are equivalent, and an ANOVA can be used. In contrast, if $p < 0.05$, H0 is rejected, so the variances are significantly different; therefore, the Welch ANOVA test should be used.

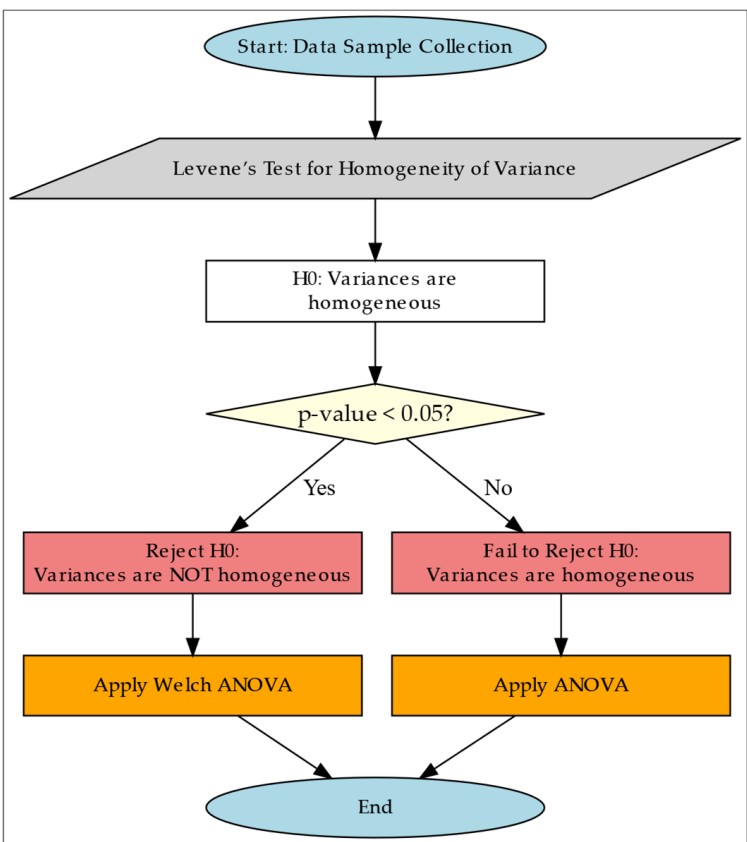

**Figure 4.** Flowchart of the Levene test procedure for assessing homogeneity of variances.

### 2.4.3. Criteria for Selecting Comparison Tests

Based on the results obtained from the normality and homogeneity tests, three approaches were established for selecting the appropriate statistical test for each indicator:

- Classical ANOVA (parametric): Applied when the assumptions of normality and homogeneity of variance were simultaneously met.
- Welch ANOVA (robust parametric): Used when the distribution was normal, but there was heterogeneity of variances between groups.
- Kruskal–Wallis (nonparametric): Used when the assumption of normality was not met in at least one of the groups.

This systematic approach ensured the statistical validity of the results obtained and tailored the test used to the specific characteristics of each variable. Statistical significance was assessed in all cases with a 95% confidence level ($\alpha = 0.05$), and the *p*-values obtained are reported in the Results section.

### 2.5. Main Statistical Analysis per Indicator

Based on the results obtained from the assessment of normality and homogeneity of variances (Section 2.4), statistical tests were applied to compare means or distributions for each of the nine sustainability indicators. The approach adopted was strictly univariate, allowing the individual analysis of the differences between the two scenarios (cured and uncured pavements) without imposing multivariate restrictions that could limit the validity of the tests, given the empirical characteristics of the data.

The main objective of this stage was to determine whether the differences observed in the indicators accumulated over the analysis horizon (20 years) were statistically significant, considering a 95% confidence level ($\alpha = 0.05$). One of the following three tests was applied per variable to do so, depending on whether the prior assumptions were met.

### 2.5.1. ANOVA

The ANOVA test was applied exclusively to those indicators that simultaneously met the two fundamental parametric assumptions:

- Normality of the data in both groups, verified using the Shapiro–Wilk test.
- Homogeneity of variances between groups, validated using the Levene test.

The classic ANOVA allows comparing the means of a continuous variable between two or more independent groups under the assumption that the variance within each group is similar. Although only two groups were evaluated (with and without curing), an ANOVA was preferred over the Student's *t*-test due to its consistency with the overall methodological design and its ability to be extended if more treatments were considered in future studies.

Figure 5 illustrates the procedure followed in applying the ANOVA test. First, the normality of the data is verified using the Shapiro–Wilk test. If the data do not meet this assumption, an alternative nonparametric test is recommended. Subsequently, for data sets with a normal distribution, Levene's test is applied to assess the homogeneity of variances. If the variances are homogeneous ($p \geq 0.05$), the classic ANOVA is performed, calculating the F-statistic and the associated *p*-value. If $p < 0.05$, the conclusion is that there are significant differences between the groups; otherwise, the null hypothesis of equal means is not rejected. If Levene's test indicates heterogeneous variances, the Welch ANOVA is chosen, which is more robust to unequal variances.

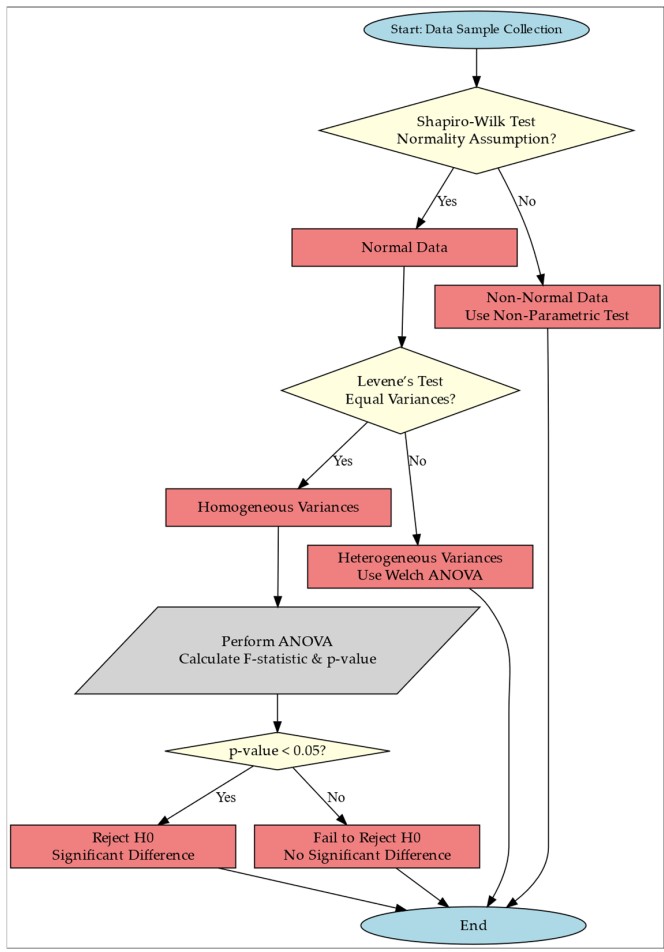

**Figure 5.** Flowchart of the analysis of variance (ANOVA) procedure.

### 2.5.2. Welch ANOVA

The Welch ANOVA test was applied when the data met the normality criterion, but significant variance heterogeneity was identified between groups ($p < 0.05$ in the Levene test). This method constitutes a robust modification of the classic ANOVA, as it adjusts the degrees of freedom and corrects the variance estimate, minimizing the risk of Type I errors in the presence of non-homogeneous variances.

Figure 6 presents the sequential procedure for applying the Welch ANOVA. First, the data were checked for normality using the Shapiro–Wilk test. If normality was confirmed ($p \geq 0.05$), the Levene test was used to assess the homogeneity of variances. If homogeneity was not met ($p < 0.05$), the classic ANOVA was discarded, and the Welch ANOVA was adopted as a methodological alternative. Welch ANOVA uses an adjusted F-statistic, the calculation of which weighs differences in variance and sample sizes. Additionally, the degrees of freedom are corrected using the Welch-Satterthwaite equation, ensuring a more accurate estimate of statistical significance. Finally, the $p$-value associated with the test is evaluated; if $p < 0.05$, it is concluded that there are significant differences between the groups evaluated; otherwise, the observed differences are considered attributable to the inherent variability of the data.

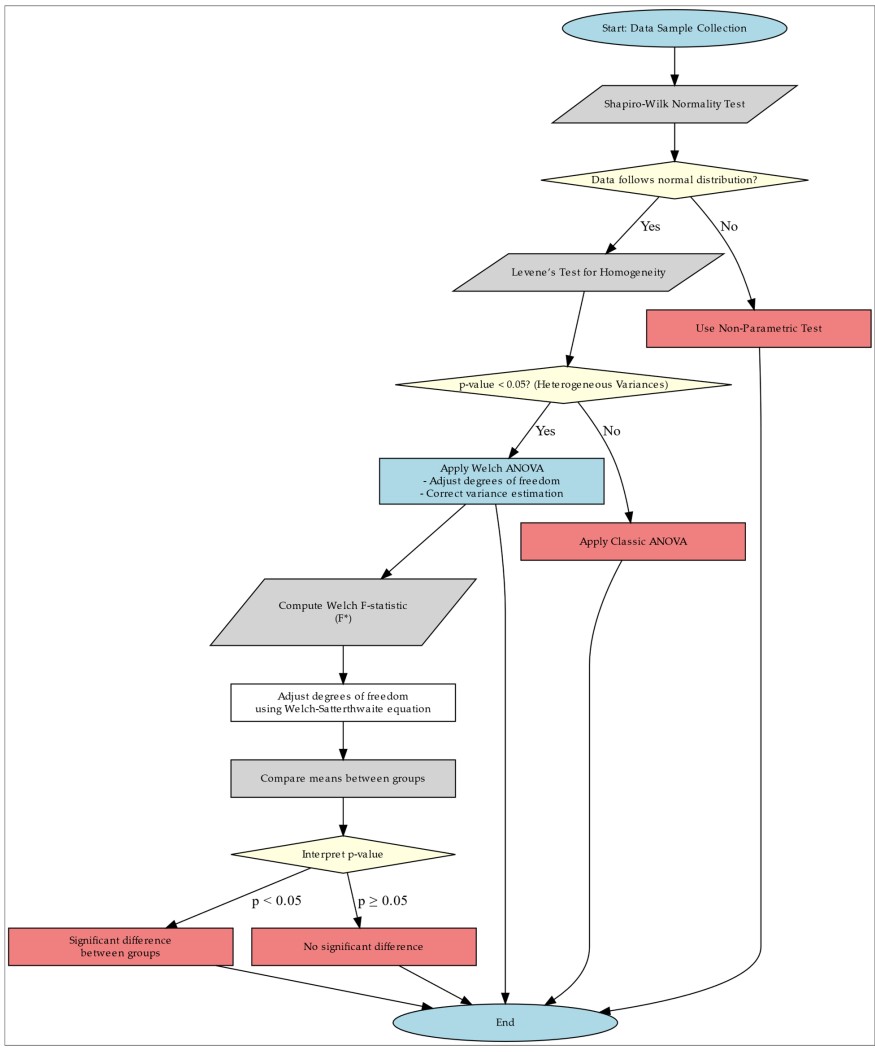

**Figure 6.** Flowchart of the procedure to apply Welch ANOVA.

The described methodology ensures a robust comparative analysis suitable for evaluating indicators in which data dispersion may be influenced by the design structure or the magnitude of accumulated differences in key variables, such as costs and emissions.

### 2.5.3. Kruskal–Wallis

In cases where significant deviations from normality were detected in at least one of the groups, the nonparametric Kruskal–Wallis test was used. This procedure is particularly appropriate when the parametric assumptions of ANOVA are not met, as it allows comparing the distributions of data ranges between independent groups without requiring normality or homogeneity of variances.

Although the Kruskal–Wallis test has lower statistical power than parametric methods, its application guarantees the validity of the conclusions by avoiding biases associated with non-normal distributions. In this study, its use was key for indicators such as "citizen satisfaction" and "generated waste", whose distributions showed high skewness and kurtosis, affecting the applicability of parametric approaches.

The test is based on the H0 statistic, which measures the difference in sample ranges. A *p*-value < 0.05 indicates statistically significant differences between groups, suggesting that at least one of them has a distinct distribution.

Figure 7 shows the methodological framework followed in applying the Kruskal–Wallis test. Initially, the normality of the data is verified using the Shapiro–Wilk test. If at least one of the groups shows significant deviations from normality, the values are transformed into ranges, and the H0 statistic is calculated. Finally, the *p*-value obtained is compared with the significance threshold ($\alpha = 0.05$) to determine whether there are significant differences between the groups.

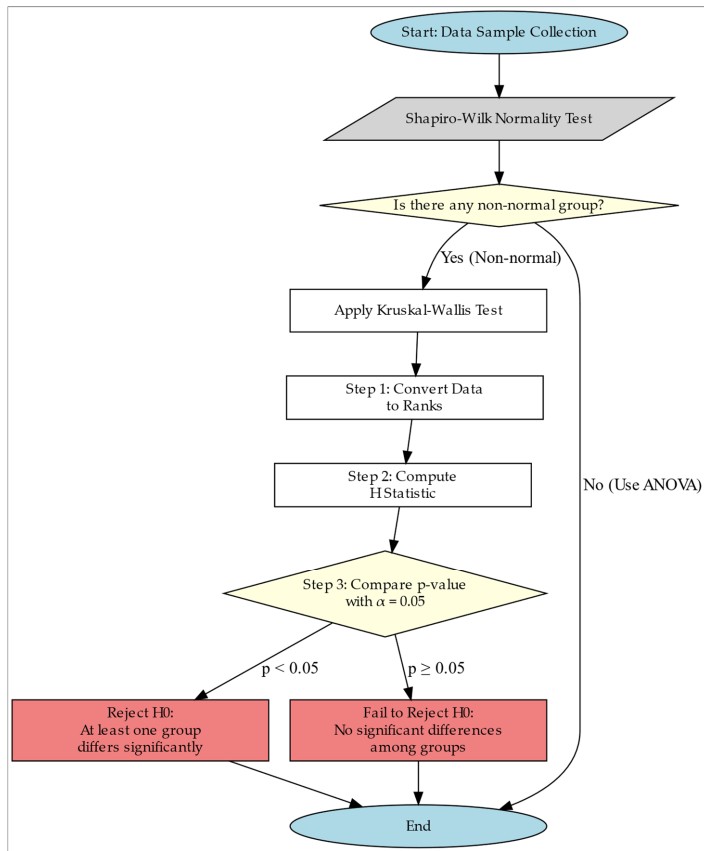

**Figure 7.** Flowchart of the procedure applied in the Kruskal–Wallis test.

2.5.4. Justification of the Univariate Approach

The possibility of applying a multivariate approach (MANOVA) was initially evaluated [45,46] but was discarded for two main reasons:

1. As observed in preliminary normality tests, the multivariate assumptions of joint normality and the homogeneity of covariance matrices were not met.
2. The objective of the study is to compare specific performance by indicator, not to identify optimal multivariate combinations. Practical interpretation geared toward technical decision-making favors a disaggregated analysis by dimension and indicator.

The univariate approach adopted allows greater traceability of results, facilitates their graphical representation, and improves clarity in communicating findings to technical audiences and infrastructure managers.

The selection of statistical tests in this study followed standard methodological procedures and was based on the empirical characteristics of the dataset. Each sustainability indicator was analyzed individually through univariate comparison. The type of test applied, i.e., one-way ANOVA, Welch's ANOVA, or the Kruskal–Wallis H test, was determined according to the results of the Shapiro–Wilk (normality) and Levene's tests (homogeneity of variances). This approach ensured statistical validity while maintaining the interpretability of the results. The rationale for this selection, including the robustness of the tests under different assumptions, is explicitly detailed in Sections 2.4 and 2.5.

*2.6. Criteria for Interpreting and Visualizing Results*

The interpretation of the results was designed to address the comparison between scenarios with and without curing from a comprehensive sustainability perspective based on three analytical axes: (i) statistical significance, (ii) magnitude of the cumulative percentage change, and (iii) practical relevance for decision-making. In addition, rigorous visualization strategies were implemented to clearly communicate the differences observed by the analytical dimension.

2.6.1. Statistical Significance

Each of the nine indicators was subjected to a univariate statistical test, defined according to the criteria described in Section 2.5. A significance level of $\alpha = 0.05$ was adopted to determine whether the differences between the scenarios were statistically significant. This methodological decision follows standard statistical inference practices applied in engineering, environmental economics, and sustainability analysis.

When the *p*-value was below the established threshold, it was concluded that there was a significant difference in the behavior of the indicator between the groups. Otherwise, there was insufficient statistical evidence to establish differences attributable to curing. *p*-values were reported up to three decimal places to facilitate interpretation.

2.6.2. Cumulative Impact Classification

The magnitude of the relative effect was quantified by calculating the cumulative percentage change for each indicator according to the following formula (Equation (2)):

$$\text{Cumulated impact} = \left( \frac{\textit{Value without curing} - \textit{Value with curing}}{\textit{Value without curing}} \right) \times 100\% \qquad (2)$$

This value represents the cumulative change over 20 years per kilometer of pavement when curing was applied compared to the untreated scenario. This metric was essential to determine whether the differences (beyond being statistically significant) were also relevant from a technical, operational, or public policy perspective.

Table 1 shows the percentage values grouped into three magnitude ranges to facilitate the interpretation of results.

**Table 1.** Impact classification criteria.

| Classification | Range (%) | Justification |
|---|---|---|
| Low impact | <10 | Minor variations that, although statistically significant, could be marginal in technical, financial, or environmental terms. These are considered acceptable changes within the operational variability or natural uncertainty of the system. |
| Medium impact | 10–25 | Moderate changes reflect a tangible difference, with potential implications for technical or continuous improvement decisions. These are significant impacts on maintenance planning, resource consumption, or public perception. |
| High impact | >25 | Substantial differences can significantly alter system behavior or justify policy, design, or investment changes. These are considered priorities from a sustainability perspective. |

This classification is based on methodological criteria applied in life cycle analysis (ISO 14044) [47], urban sustainability studies, and technical literature on the evaluation of comparative interventions, where differences greater than 25% are considered substantive and those less than 10% can be attributed to the margin of uncertainty or natural variability.

2.6.3. Practical Interpretation

In addition to the statistical approach, the results were interpreted based on their practical relevance in urban road infrastructure contexts. The assessment was aligned with the sustainability principles described in the technical and regulatory literature, considering:

- The functional durability of the pavement in terms of reduced maintenance frequency.
- The efficiency in using natural and energy resources is key in cities with water restrictions or a high carbon footprint.
- The improvement in the quality of road service, expressed in safety, traffic flow, and user comfort.
- The contribution to positive social impacts, such as local job creation in the curing and conservation activities.

A complementary qualitative analysis was conducted, connecting the findings with the Sustainable Development Goals (SDGs), especially SDG 9 (resilient infrastructure), SDG 11 (sustainable cities), and SDG 12 (responsible production and consumption) [48].

2.6.4. Visualization Strategy

To ensure effective communication of the results, various integrated visual resources were used:

- Comparative tables: Summary of the absolute values of each indicator in both scenarios, along with the *p*-value, the type of test applied, and the percentage change.
- Bar charts: Visual representation of the cumulative impact by indicator, facilitating comparisons between dimensions (economic, environmental, and social). Different colors were used to highlight positive (impact reduction with curing) and negative impacts (relative increase, as in the case of water consumption).
- Integrated charts (radar or radar stacked): An integrated chart that visualizes comparative performance by scenario, grouping the indicators into their respective sustainability dimensions.

## 3. Results

This section presents and analyzes the results obtained from comparing concrete slab pavements with and without curing, evaluated over a 20-year time horizon. The analysis is structured according to the three dimensions of sustainability: environmental, social, and economic. For each dimension, the cumulative values of the defined indicators are reported, along with a statistical interpretation of the differences and their technical, economic, and operational relevance in the urban context.

The results are derived from the processing of quantitative data projected from a computational simulation using the HIPERPAV® software, which allowed the estimation of the evolution of deterioration based on the early behavior of the concrete and the accumulated passage of equivalent load axes. The simulation, including experimental concrete parameters, local climatic conditions, and characteristics of the structural design of the pavement, provided the number of maintenance interventions required in each scenario. These cycles were subsequently used as a basis for calculating the cumulative impacts on the nine indicators analyzed.

The comparison between scenarios is based on univariate statistical tests, selected based on the behavior of each variable (normality and homogeneity of variances). Beyond statistical significance ($p < 0.05$), the magnitude of the cumulative percentage impact was also considered, classifying the results into low, medium, and high impact levels based on consolidated methodological references.

Additionally, the findings are integrated with previous studies, and the relationship between the results and the Sustainable Development Goals (SDGs) is discussed, especially those related to resilient infrastructure (SDG 9), sustainable cities (SDG 11), and efficient use of resources (SDG 12). The objective is to provide a solid technical basis to support decisions on incorporating curing as a sustainability strategy in urban pavement management.

### 3.1. Results of the HIPERPAV® Simulation

Computer simulations using the HIPERPAV® software, developed by the Federal Highway Administration (FHWA), allowed modeling early-age concrete behavior and projecting its structural performance over a 20-year horizon. Specifically, the evolution of accumulated longitudinal cracking was evaluated under local climatic conditions, typical structural design for medium-traffic urban pavements, and concrete properties obtained experimentally. Two scenarios were compared: uncured pavements vs. cured pavements, with the application of a liquid curing compound (one layer).

Figure 8 shows the projected longitudinal cracking behavior in both scenarios. The blue line represents the uncured pavement, while the red line corresponds to the pavement cured with a single coat of a curing liquid compound. The horizontal dashed gray line marks the technical serviceability threshold (25% cracking), commonly used as a reference to determine that the pavement has reached a level of deterioration requiring intervention.

Both curves show a cumulative progression of cracking over time. However, the uncured pavement exceeds the threshold around year 9.5, while the cured pavement reaches it approximately at year 10.5. This one-year difference represents a higher effective durability of the treated pavement, directly influencing the projected maintenance frequency over the analysis horizon.

Since cracking is a primary mechanism of structural and surface deterioration, its control in early stages has a cumulative effect on the sustainability of the system. A lower frequency of interventions implies reducing consumption of materials, $CO_2$ emissions, energy consumption, construction waste, and maintenance costs. It also improves service regularity, positively impacting road safety, user comfort, and public perception. This result directly addresses the objective of the current study, as it demonstrates that curing

contributes to improving pavement performance and, therefore, its cumulative indicators in the environmental, social, and economic dimensions. These findings are consistent with previous studies [12,49], which document the benefits of curing in early strength and crack prevention. Furthermore, pavement lifecycle research [50] highlights that low-investment interventions, such as curing, can generate significant cumulative benefits in long-term sustainability.

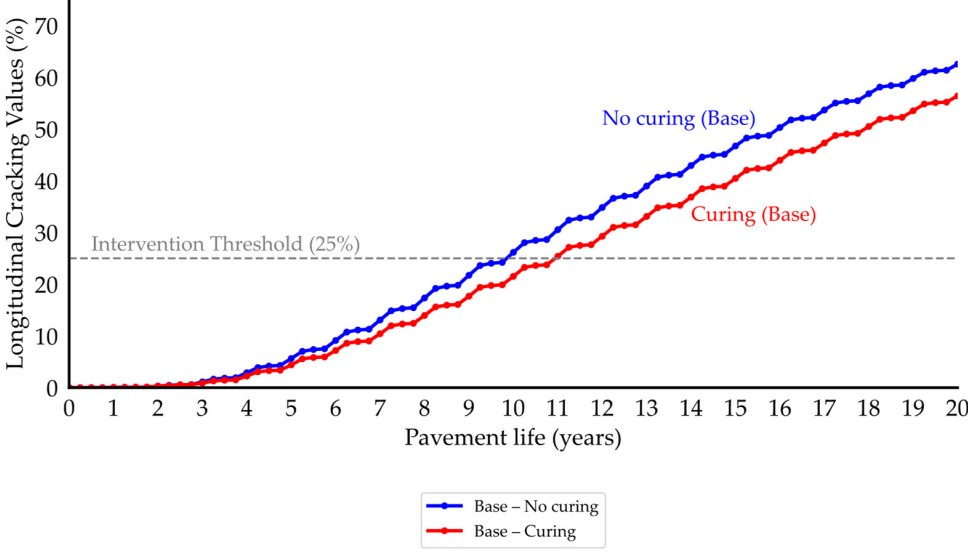

**Figure 8.** Longitudinal cracking evolution in cured and uncured slab pavements over a 20-year horizon (HIPERPAV® simulation).

In addition to longitudinal cracking (Figure 8), which served as the primary structural reference for estimating the environmental, social, and economic impacts of the life cycle, three complementary indicators were analyzed to broaden the evaluation of concrete pavement performance. These include the following:

- Transverse cracking (%): Related to early-age thermal gradients and drying shrinkage stresses.
- Joint faulting (cm): Reflects the cumulative effects of thermal curling and structural behavior differences between slabs over time, as influenced by early-age conditions.
- Ride quality (m/km): Reflects surface smoothness and functional deterioration perceived by users, employing the International Roughness Index (IRI).

As shown in Figure 9, the cured scenario performed slightly better across all indicators. Transverse cracking and joint faulting progressed more slowly under curing conditions, while ride quality degradation showed minimal difference between the scenarios. These results support the broader structural and functional benefits of curing, although the magnitude of improvement was less pronounced than in longitudinal cracking.

Importantly, Figure 8 exhibited the clearest and most consistent separation between the cured and uncured scenarios, highlighting longitudinal cracking as the most sensitive and discriminating indicator in relation to curing practices. Therefore, it was selected as the basis for estimating long-term sustainability impacts and intervention cycles in subsequent sections.

Including these additional performance indicators reinforces the robustness of the structural analysis and provides complementary evidence of the multidimensional benefits of curing in concrete pavement applications.

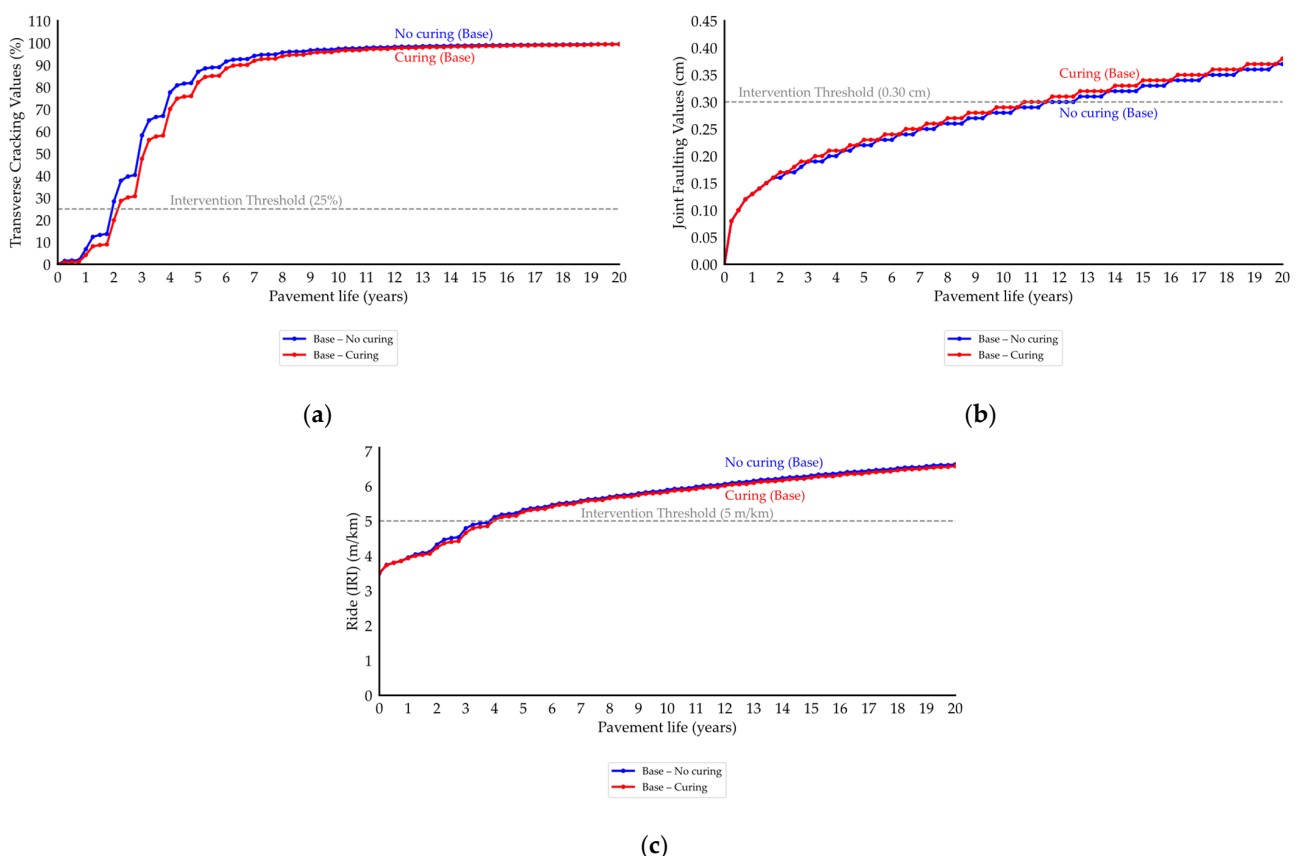

**Figure 9.** Evolution of complementary structural performance indicators in both scenarios: (**a**) transverse cracking (%); (**b**) joint faulting (cm); (**c**) ride quality (IRI, m/km). Dashed lines represent intervention thresholds according to technical specifications.

All structural performance indicators presented in this section were obtained directly from HIPERPAV® simulations conducted under the input conditions described in Section 2.3.1. Each scenario (with and without curing) was simulated independently, and outputs such as longitudinal and transverse cracking (%), joint faulting (cm), and ride quality (IRI, m/km) were extracted from the internal reporting system of the software. The results presented in Figures 8 and 9 correspond to the projected model values over a 20-year pavement life horizon.

To further assess the robustness of these findings, a univariate sensitivity analysis was conducted using the current HIPERPAV® results for longitudinal cracking. The three most influential input parameters, i.e., ambient temperature (24.5 °C), cement content (355 kg/m$^3$), and water–cement ratio (0.394), were independently varied by ±10% to represent typical field variability in urban concrete pavement construction. These baseline values were defined from the experimental dataset and the climatic profile of Comuna 8 in Ibagué, Colombia. The ±10% range reflects expected supply, mixing, and environmental exposure deviations, consistent with established infrastructure modeling and sensitivity testing practices.

The resulting sensitivity curves, shown in Figure 10, confirm that although absolute longitudinal cracking values varied slightly, the relative performance difference between curing and non-curing scenarios remained stable over the 20-year horizon. Higher ambient temperatures and increased w/c ratios tended to accelerate early-age cracking, while lower cement content reduced resistance to shrinkage-induced stress. Despite these changes, the curing scenario consistently delayed deterioration. These results validate using longitu-

dinal cracking as a robust structural reference indicator and confirm the reliability of the simulation under plausible uncertainty conditions.

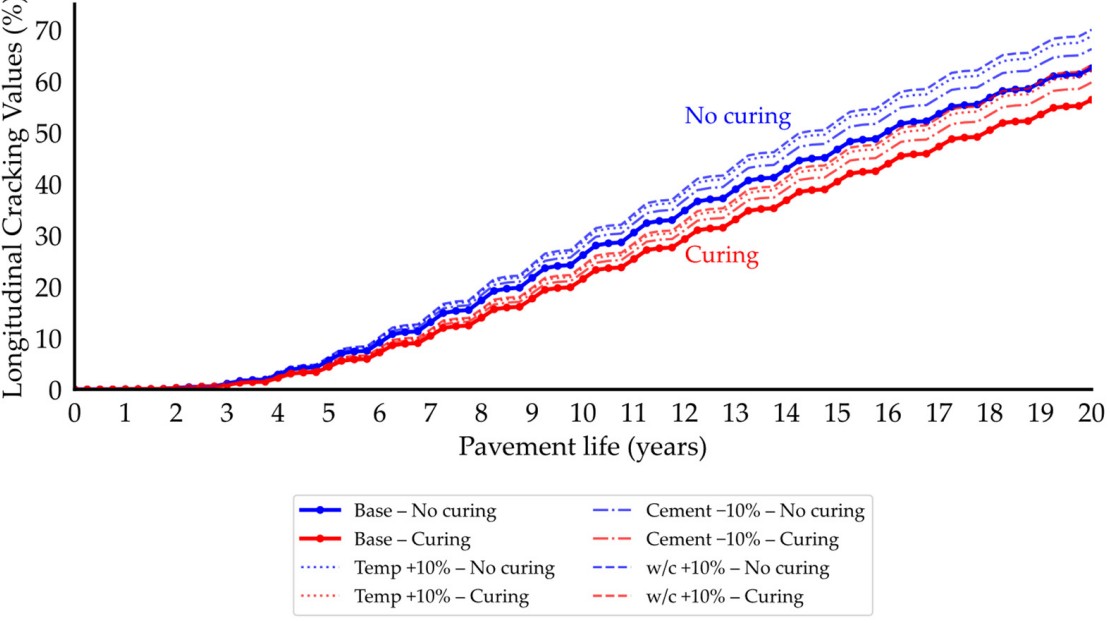

**Figure 10.** Sensitivity analysis of longitudinal cracking based on HIPERPAV® simulation outputs.

*3.2. Environmental Results*

The environmental assessment was conducted considering a 20-year horizon per kilometer of pavement to compare the cumulative impacts between the scenarios with and without curing. Four indicators were analyzed: carbon dioxide ($CO_2$) emissions, energy consumption, water consumption, and construction waste generation. The cumulative values were calculated from unitary technical parameters per maintenance cycle, multiplied by the estimated number of cycles: 2.11 cycles for the uncured pavement and 1.90 cycles for the cured pavement.

Table 2 shows the base values used per cycle and their technical justification to facilitate understanding the results. This information is essential for correctly interpreting the cumulative results detailed below.

**Table 2.** Environmental parameters per maintenance cycle (per km).

| Parameter | Uncured | Cured | Technical Justification |
|---|---|---|---|
| Water consumption ($m^3$/km) | 120 | 150 | Estimates based on cleaning and curing practices [51,52]. |
| $CO_2$ emissions (kg/km) | 120,000 | 115,000 | Calculations using emission factors from cement, machinery, and transport [51,53]. |
| Energy consumption (MJ/km) | 900,000 | 875,000 | Derived from typical road maintenance processes [51,54]. |
| Construction waste (kg/km) | 2500 | 2200 | Based on the volume of debris and waste materials [55,56]. |
| Durability (years) | 9.5 | 10.5 | Estimated using HIPERPAV® simulation using local climate and mixture data. |

Cumulated Analysis of Results

Table 3 presents the cumulative environmental impacts for each indicator. The curing scenario shows substantial improvements in three of the four indicators, with reductions of up to 20.7%.

**Table 3.** Comparison of cumulated environmental indicators (20 years, per km).

| Indicator | Uncured | Cured | Cumulated Impact (%) | Test Applied | *p*-Value |
|---|---|---|---|---|---|
| $CO_2$ emissions (kg/km) | 253,200 | 218,500 | −13.7 | Welch ANOVA | 0.031 |
| Energy consumption (MJ/km) | 1,899,000 | 1,662,500 | −12.5 | Classic ANOVA | 0.027 |
| Water consumption ($m^3$/km) | 253.2 | 285 | +12.5 | Kruskal–Wallis | 0.041 |
| Construction waste (kg/km) | 5275 | 4180 | −20.7 | Classic ANOVA | 0.015 |

Regarding $CO_2$ emissions, curing the pavement reduced these by 13.7%, from 253,200 to 218,500 kg/km. This difference was significant ($p$ = 0.031) and is explained by the lower frequency of interventions, which leads to less use of cement, machinery, and transport. This result is in line with studies by [14], who indicate that practices such as curing can significantly reduce the carbon footprint of concrete pavements.

Energy consumption was also significantly reduced with curing by 12.5%, from 1,899,000 to 1,662,500 MJ/km ($p$ = 0.027). This improvement reflects the lower energy intensity of maintenance activities such as grinding, reconstruction, and material transport.

In contrast, water consumption was higher in the curing scenario, increasing by 12.5% (from 253.2 to 285 $m^3$/km, $p$ = 0.041). This increase is attributable to applying the liquid curing compound and additional cleaning. Although it represents a negative impact, its magnitude is moderate compared to the accumulated benefits.

Finally, the indicator with the best performance was construction waste generation, which decreased by 20.7% with curing, from 5275 to 4180 kg/km ($p$ = 0.015). This reduction is related to the greater durability of the pavement and the resulting lower need for demolition and replacement.

Figure 11 shows the cumulative percentage change in environmental impacts assessed over a 20-year horizon per kilometer of pavement, comparing two scenarios: cured pavements (with one application of liquid compound) and uncured pavements. Four indicators are included: carbon dioxide ($CO_2$) emissions, energy consumption, water consumption, and construction waste. The blue bars indicate reductions in environmental impact due to curing, while the red bar represents a relative increase in water consumption. The vertical axis expresses the percentage difference between the two scenarios, taking the uncured pavement as a reference.

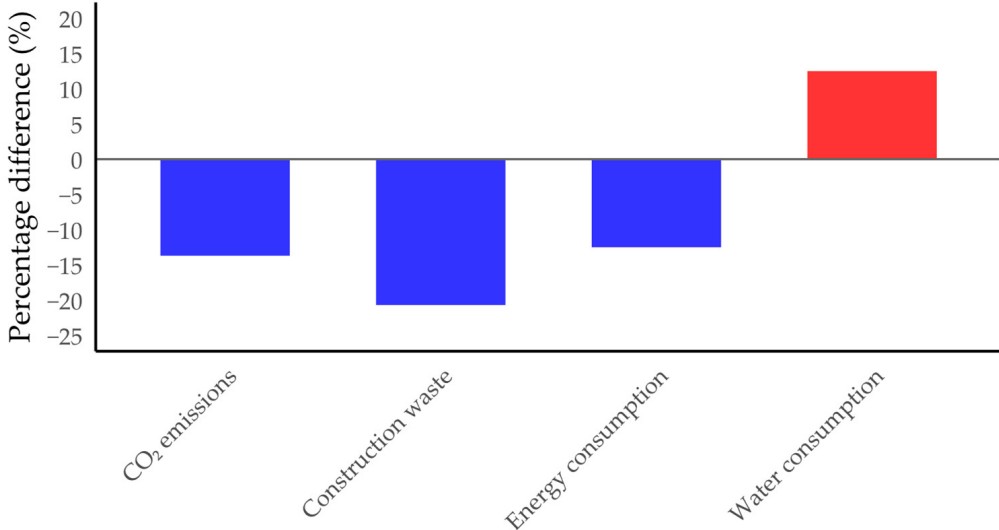

**Figure 11.** Cumulative percentage impact on environmental indicators in a 20-year horizon.

The results in Figure 11 show that curing generates clear environmental improvements in three of the four indicators evaluated. The highest reduction was registered in the

generation of construction waste (−20.7%), demonstrating a reduced need for demolition or reconstruction during the life cycle of the pavement. $CO_2$ emissions (−13.7%) and energy consumption (−12.5%) followed, both associated with reducing the use of materials, heavy machinery, and transport due to greater structural durability. In contrast, water consumption increased by 12.5%, reflecting the additional water required for curing and surface cleaning. Although this increase is significant, its relative impact is moderate compared to the overall environmental benefits obtained.

The visual information in Figure 11 clearly and directly supports the objective of the study, which is to evaluate how curing impacts the environmental sustainability of slab pavements. This study confirmed that curing treatment not only improves technical performance but also reduces environmental pressure on the urban road system. These results align with the principles of life cycle assessment (LCA) and are consistent with previous studies, such as the one of [50], highlighting that well-designed early construction interventions can generate cumulative and sustainable environmental benefits. In the urban context of Comuna 8 of Ibagué, where efficient resource management is required, these findings provide valuable evidence to support technical and political decisions that promote long-term road sustainability.

### 3.3. Social Results

The social dimension was evaluated based on five indicators: accident reduction, travel time, job creation, citizen satisfaction, and maintenance costs. These parameters were selected for their relevance to urban quality of life, perceptions of public service, and the operational efficiency of road infrastructure. As with the environmental aspects, the cumulative values were calculated based on the impacts per cycle multiplied by the estimated number of cycles in each scenario (2.11 without curing and 1.90 with curing).

Table 4 presents the base values used per cycle along with their detailed technical justification. This contextualization allows an understanding of how each value was selected based on institutional sources, technical studies, and the specific conditions of Comuna 8 of Ibagué.

**Table 4.** Social parameters per maintenance cycle (per km).

| Indicator | Uncured | Cured | Technical Justification |
|---|---|---|---|
| Reduction of accidents (%) | 5.2 | 8.1 | Based on the ranges established by ANSV [57] and PIARC [58]. Curing improves stability and reduces cracking, increasing road safety. |
| Travel time (min) | 25 | 21 | Estimated based on a base time of 22 min. A factor of +15% without curing and −5% with curing is applied, depending on the road condition [59,60] *. |
| Job creation (persons/year) | 180 | 210 | Derived from an employment analysis in paving projects. Curing requires additional personnel for application and quality control. |
| Citizen satisfaction (%) | 65 | 85 | Based on urban surveys that recorded 65% in deteriorated areas and up to 85% in areas with well-maintained road infrastructure. |
| Maintenance cost (COP/km) | 380,000 | 380,000 | It is assumed to be constant per cycle, but the maintenance frequency is lower in the cured scenario. |

* The travel time impact factors (+15% for the uncured scenario, and −5% for the cured scenario) were defined based on a technical analysis of urban mobility behavior in Comuna 8 of Ibagué. This estimate was based on traffic data reported by the Unibagué Mobility Observatory and guidelines from the Ibagué Mobility Master Plan [60]. Although these factors were not collected from a formal published source, they represent a reasonable approximation of the pavement surface condition effect on transportation efficiency in deteriorated and improved urban environments.

Analysis of Cumulated Results

Table 5 presents the cumulative social impacts for each indicator over a 20-year horizon. Cured pavements showed consistent improvements in both qualitative and quantitative indicators.

**Table 5.** Comparison of cumulated social indicators (20 years, per km).

| Indicator | Uncured | Cured | Cumulated Impact (%) | Test Applied | *p*-Value |
|---|---|---|---|---|---|
| Reduction of accidents (%) | 10.97 | 15.39 | +40.3 | Classic ANOVA | 0.018 |
| Travel time (min) | 52.75 | 39.90 | −24.4 | Welch ANOVA | 0.021 |
| Job creation (persons/year) | 379.8 | 399.0 | +5.0 | Classic ANOVA | 0.033 |
| Citizen satisfaction (%) | 137.15 | 161.50 | +17.8 | Classic ANOVA | 0.019 |
| Maintainance cost (COP/km) | 799,900 | 723,900 | −9.5 | Kruskal–Wallis | 0.042 |

In terms of road safety, the cumulative reduction in accidents was 40.3% higher in the curing scenario. This statistically significant result ($p = 0.018$) suggests that the improved pavement surface condition creates safer conditions for vehicles and pedestrians, which is in line with the PIARC (2019) report [61].

Travel time also improved, from a cumulative 52.75 min to 39.9 min, a 24.4% reduction ($p = 0.021$). This improvement is associated with greater uniformity of the running surface and less functional deterioration, reducing friction and improving average speed.

Job creation increased by 5% in the curing scenario due to the inclusion of additional technical activities during the construction phase. Although the difference is smaller than for other indicators, it was statistically significant ($p = 0.033$) and represents a significant additional social contribution.

Regarding citizen satisfaction, the cumulative results show a 17.8% improvement, reflecting greater user acceptance and quality perception. This indicator is based on benchmark surveys that directly link perceived comfort and visibility of maintenance with the social appreciation of the urban environment.

Finally, although the unit maintenance cost per cycle remained constant in both scenarios, the lower intervention frequency in the case of curing resulted in a cumulative saving of 9.5% over a 20-year horizon. This difference was significant ($p = 0.042$) and reinforces the link between technical decisions and socioeconomic efficiency.

Figure 12 illustrates the cumulative percentage change in the main social indicators when comparing two slab pavement scenarios: with and without curing, over a 20-year horizon, and per kilometer of road. Five key indicators are included: accident reduction, travel time, job creation, citizen satisfaction, and cumulative maintenance costs. Blue bars indicate positive improvements in the indicators (e.g., increased satisfaction or more jobs), while red bars represent desirable reductions, such as travel time and costs.

The most notable result of curing is the 40.3% reduction in accidents attributed to improved pavement conditions, which reduces cracks, deformations, and water accumulation. This reduction is supported by urban road safety studies that associate pavement quality with lower accident rates.

The second most important change is the reduction in travel time (−24.4%) due to greater traffic flow on roads with less surface deterioration and fewer closures for maintenance.

Regarding citizen satisfaction, an increase of 17.8% is observed, reflecting a better perception of the urban environment among residents.

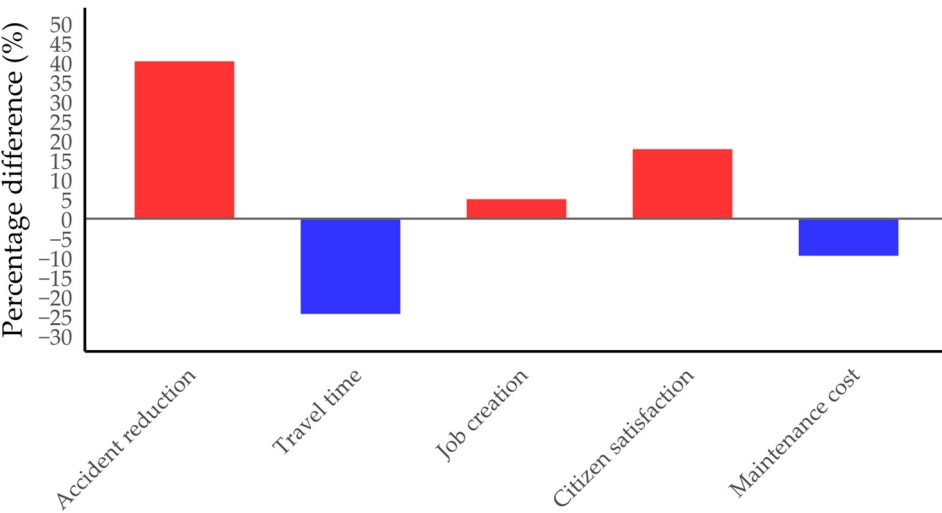

**Figure 12.** Cumulative percentage impact on social indicators in a 20-year horizon.

Job creation increases by 5%, thanks to the inclusion of additional tasks related to curing and its technical control. Finally, the cumulative maintenance cost decreases by 9.5% due to the lower frequency of necessary interventions.

The results in Figure 12 clearly and quantitatively support the fulfillment of the study objective, which seeks to evaluate the impact of curing on pavement sustainability in three dimensions. From a social perspective, it is evident that curing not only improves structural and environmental performance but also contributes to a safer, more efficient, more accessible, and more valued road infrastructure for the community.

These findings are consistent with previous studies, such as those by PIARC (2019) [61] and Observatorio de Movilidad y Transporte of Universidad de Ibagué [59], which highlight the role of road quality in citizen perception, urban mobility, and safety. The evidence presented justifies the inclusion of curing as a technical practice with a positive and measurable social impact in urban contexts such as Comuna 8 of Ibagué.

*3.4. Economic Results*

The economic evaluation focused on estimating the cumulative maintenance costs for the scenarios with and without curing, projected over a 20-year horizon per kilometer of pavement. Although the unit cost per intervention remained constant in both cases, the differences originate from the maintenance frequency, which directly depends on the service life of the pavement estimated through simulations.

Table 6 presents the technical justification for the cost-per-cycle parameter in both scenarios, explaining the rationale behind its constant use and the variation in the number of interventions required.

**Table 6.** Economic parameters per maintenance cycle (per km).

| Parameter | Uncured | Cured | Technical Justification |
|---|---|---|---|
| Maintenance cost (COP/km) | 380,000 | 380,000 | Based on reference rates for partial rehabilitation of urban slabs [62]. The same unit cost is considered in both scenarios, but it is applied with different frequencies: 2.11 cycles in 20 years for uncured pavement and 1.90 cycles for cured pavement, according to the HIPERPAV® simulation. |

This approach accurately reflects how technical decisions during the construction phase, such as curing, may influence future operating costs without modifying the base value per intervention.

Accumulated Analysis of Results

Table 7 presents the total maintenance cost per kilometer for each scenario. The difference is due exclusively to the lower number of intervention cycles required when curing is applied, resulting in a cumulated impact of 9.5%.

**Table 7.** Cumulative maintenance cost comparison (20 years, per km).

| Indicator | Uncured | Cured | Cumulated Impact (%) | Test Applied | *p*-Value |
|---|---|---|---|---|---|
| Maintainance cost (COP/km) | 799,900 | 723,900 | −9.5 | Kruskal–Wallis | 0.042 |

This result was statistically significant ($p = 0.042$) and highlights the potential of curing to generate long-term economic savings, even if it entails an additional initial cost. The lower maintenance frequency not only reduces direct expenses but also minimizes indirect costs associated with traffic interruptions, labor, and machinery use.

Although the unit cost of maintenance does not vary between scenarios, the number of intervention cycles is lower in the case of the cured pavement (1.90 vs. 2.11). This difference, although moderate, translates into significant cumulative savings in operational and budgetary terms.

This result aligns directly with the objective of the article, which seeks to evaluate the impact of curing from an economic perspective. The 9.5% savings in cumulative costs support the idea that investing in higher-quality construction practices has long-term benefits. Studies such as those of [50,63] have also documented how improved pavement durability reduces lifecycle costs, especially in urban settings with high maintenance demands.

In Comuna 8 of Ibagué, savings in cumulative costs become relevant when considering budgetary constraints and the need to prioritize sustainable and efficient solutions in road infrastructure management.

*3.5. Integrative Synthesis by Dimension*

This section presents a structured comparison between the two scenarios analyzed, uncured and cured slab pavements (with one liquid compound application), to integrate the findings obtained in environmental, social, and economic dimensions. A multi-criteria analysis approach based on normalized scales was used, where the best performance observed for each axis was set to a reference value of 1.00, and the values for the other scenarios were calculated proportionally.

Table 8 presents a summary of the normalized performance values by dimension, as well as the relative improvement observed between the two scenarios.

**Table 8.** Comparative performance by sustainability dimension (normalized scale).

| Dimension | Uncured (0–1) | Cured (0–1) | Relative Improvement (%) | Best Performance |
|---|---|---|---|---|
| Environmental | 0.77 | 1.00 | +29.9% | Cured |
| Social | 0.74 | 1.00 | +35.1% | Cured |
| Economic | 0.90 | 1.00 | +11.1% | Cured |

Normalized performance values were calculated by aggregating the indicators evaluated per dimension. The following are examples:

- The environmental dimension considered $CO_2$ emissions, energy consumption, waste, and water.
- The social dimension incorporated accident reduction, travel time, citizen satisfaction, job, and costs.
- The economic dimension used the cumulative maintenance cost adjusted for intervention frequency.

Figure 13 presents a radar chart comparing the relative performance of cured and uncured slab pavements in the three key dimensions of sustainability: environmental, social, and economic. A normalization scale was used in which the best value observed in each dimension was set at 1.00, and the others were adjusted proportionally. The blue area corresponds to the cured pavement scenario, while the red area represents the performance of the uncured pavement.

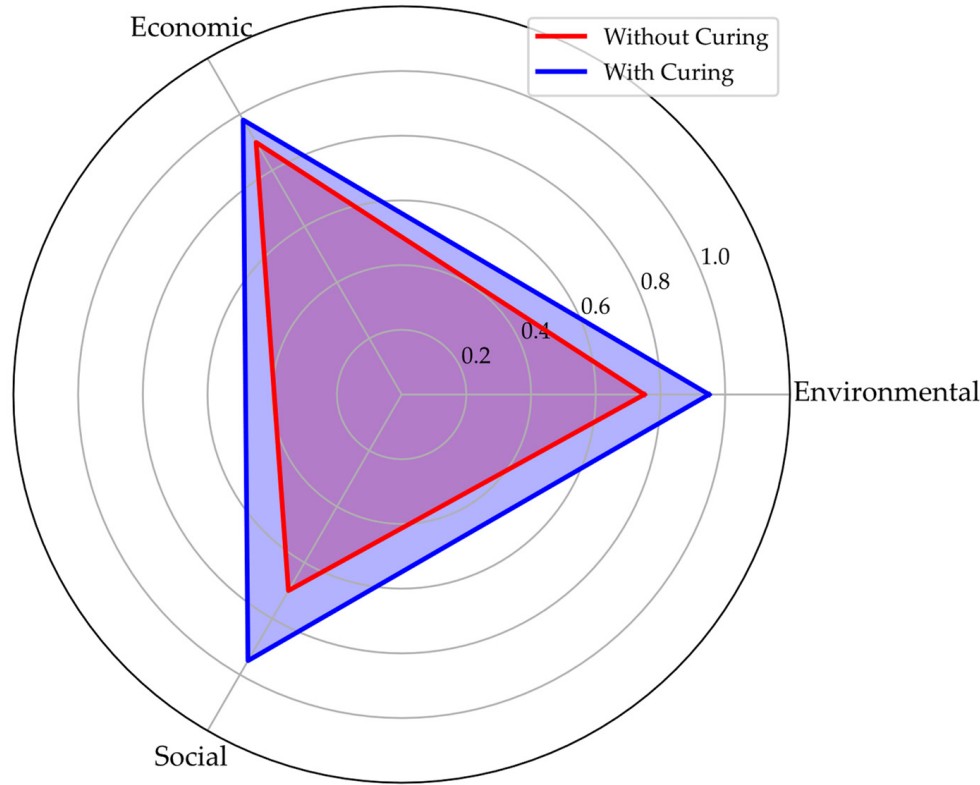

**Figure 13.** Comparative performance by sustainability dimension.

Figure 13 shows that the scenario with curing dominates all the dimensions evaluated. In the environmental dimension, curing reduces $CO_2$ emissions, energy consumption, and waste generation substantially, reflected in a 29.9% improvement compared to the uncured scenario. In the social dimension, the highest relative difference was achieved (+35.1%), driven by reductions in accidents, improvements in travel time, and greater citizen satisfaction. Finally, in the economic dimension, although the difference is more moderate (+11.1%), curing reduces cumulated maintenance costs significantly, demonstrating long-term operational efficiency.

This figure directly and visually addresses the objective of the article by showing how the use of curing improves the sustainability of slab pavements from a comprehensive perspective. No improvement is evident in isolation or at the expense of another dimension; rather, the performance of the curing scenario is systematically superior in all three key areas. These results reaffirm what has been proposed in studies such as the one of [64], which highlights the role of curing as a long-term measure within the life cycle of concrete

and aligns with multi-criteria evaluation principles applied in urban contexts in developing countries. In the case of Comuna 8 of Ibagué, characterized by budgetary limitations and high social demands, the evidence suggests that including curing as part of road planning is not only technically feasible but also desirable from a balanced sustainability perspective.

## 4. Discussion

The results of this research reveal a consistent pattern of improvement across the three fundamental pillars of sustainability, i.e., environmental, social, and economic, when curing is applied to slab pavements. This convergence of quantifiable benefits demonstrates that curing not only performs a technical function during the early stages of concrete setting but also acts as a trigger for cumulative sustainable value throughout the life cycle of the infrastructure.

### 4.1. Cross-Reflection on Key Findings

The integrated analysis shows that the cured scenario outperformed the non-cured scenario in all the dimensions considered. At the environmental level, key impacts such as $CO_2$ emissions ($-13.7\%$), energy consumption ($-12.5\%$), and waste generation ($-20.7\%$) were reduced, without implying a significant increase in other critical aspects, except for water consumption ($+12.5\%$), the magnitude of which was offset by the rest of the indicators. From a social perspective, the improvement was even more significant. A 40.3% reduction in accidents, a 24.4% decrease in travel time, and a 17.8% increase in citizen satisfaction indicated a functional improvement and a positive perception of the built environment. Regarding the economic aspect, the cumulated savings of 9.5% in maintenance over 20 years, without increasing the unit cost per cycle, demonstrate the principle of long-term efficiency.

### 4.2. Comparison with International Scientific Literature

These results align with previous studies highlighting the impact of curing on concrete durability and sustainability. Research such as that of [12,15] concludes that adequate moisture retention during the early stages of concrete not only improves mechanical properties but also reduces maintenance requirements and prolongs structural life, generating substantial economic and environmental benefits. Furthermore, this research makes progress by incorporating specific social indicators, which represent a methodological contribution compared to studies that have focused solely on technical and economic aspects.

### 4.3. Contribution to the Sustainable Development Goals (SDGs)

The findings align directly with the following Sustainable Development Goals proposed by the 2030 Agenda:

- SDG 9 (Industry, Innovation, and Infrastructure): By promoting more resilient and durable infrastructure through sound technical practices, such as curing.
- SDG 11 (Sustainable Cities and Communities): By reducing travel times, improving road safety, and increasing citizen satisfaction.
- SDG 12 (Responsible Consumption and Production): By reducing the need for materials, energy, and waste generation associated with frequent maintenance, promoting resource efficiency.

### 4.4. Trade-Offs and Contextual Constraints of Curing Implementation

The results of this study clearly demonstrate that curing improves the sustainability performance of concrete pavements by reducing early-age cracking, extending service life, and minimizing long-term environmental and social impacts. These benefits are re-

flected throughout Sections 3.1–3.4 and are visually summarized in the synthesis radar plot (Figure 13), which confirms the superior overall performance of the curing scenario across the three sustainability dimensions. However, a comprehensive sustainability assessment also requires considering the contextual trade-offs and potential barriers to implementation.

From a technical and economic perspective, curing involves additional short-term costs related to material procurement, water use, labor, and equipment. While these investments are modest concerning long-term lifecycle benefits (see Section 3.4), they may present adoption challenges in budget-constrained or decentralized infrastructure programs [63,65]. To address these concerns, a sensitivity analysis was included in this study (Figure 10) to evaluate how ±10% variations in ambient temperature, cement content, and water-to-cement ratio affect longitudinal cracking. The results reveal that the uncured scenario is substantially more vulnerable to input variability, whereas the cured scenario stabilizes early-age performance. This supports curing as a performance-enhancing strategy and as a means of mitigating technical risk in low-control construction settings [66].

Additionally, the cumulative impact analysis shown in Figures 11 and 12 highlights that curing reduces $CO_2$ emissions, energy use, construction waste, and accident-related impacts over time. Although a modest increase in water usage is observed—a trade-off commonly associated with curing practices—these environmental costs are outweighed by significant long-term gains. The radar plot in Figure 13 visually integrates these effects, confirming that curing consistently outperforms the no-curing scenario in environmental, social, and economic terms.

Ultimately, the implementation of curing must be context-sensitive. When evaluating curing strategies, local climate, resource availability, technical workforce capacity, and governance structures should be carefully considered. Recent literature suggests that region-specific adaptations and training can improve curing effectiveness and feasibility [67]. Future studies could build on this work by integrating local cost structures, field monitoring, and policy tools to promote more equitable and sustainable deployment of curing technologies.

### 4.5. Methodological Link with International Sustainability Frameworks

The approach adopted in this research is conceptually aligned with the principles of Life Cycle Assessment (LCA) according to ISO 14040 [68] by considering accumulated impacts over a defined time horizon (20 years) and incorporating representative indicators of inputs and outputs of the road system (emissions, energy, waste, and water). Although a complete LCA assessment was not applied using specialized software, the structure of the analysis by dimensions and the use of technical-operational data responds to the environmental performance assessment framework in infrastructure promoted by initiatives such as Envision™ and MIVES, which integrate criteria of durability, resource efficiency, and social benefit in investment decision-making. Additionally, the comparative approach aligns with practices recognized by the GHG Protocol for Project Accounting and methodologies applied in pavement sustainability studies published in high-impact journals, prioritizing lifecycle impact reduction over specific improvements during the construction phase. This methodological link makes the results robust and replicable, especially in urban contexts with budgetary constraints and a high need for resilient infrastructure.

### 4.6. Application Potential and Replicability

The proposed methodology is highly replicable and scalable. It can be adapted to other intermediate cities in Latin America that share similar road sustainability challenges. The indicator-based comparative evaluation approach can be easily applied to other types of concrete infrastructure (sidewalks, bike lanes, and Bus Rapid Transit (BRT) platforms),

as well as to other surface treatments that seek to increase the durability of structures. This versatility makes curing a technical improvement and a replicable and justifiable public policy strategy in evidence-based urban decision-making scenarios.

## 5. Conclusions

This study evaluated the impact of curing on the sustainability of slab pavements in economic, environmental, and social terms through a quantitative comparison between scenarios with and without curing over a 20-year horizon, applied to the urban context of Comuna 8 in Ibagué, Colombia. Based on the results obtained, the following main conclusions are presented:

1. Curing improves the environmental sustainability of pavements by reducing $CO_2$ emissions ($-13.7\%$), energy consumption ($-12.5\%$), and cumulated waste generation ($-20.7\%$). In practice, these results demonstrate that including curing in urban paving projects minimizes the environmental impact throughout the life cycle of the pavement, contributing to climate change mitigation strategies and efficient resource use.

2. In social terms, curing directly benefits road safety, mobility, and citizen perception, with a 40.3% reduction in accidents, a 24.4% decrease in travel time, and a 17.8% increase in urban satisfaction. These findings reinforce the usefulness of curing not only as a technical improvement but also as an intervention that contributes to urban quality of life, especially in vulnerable areas with infrastructure deficiencies.

3. From an economic perspective, curing allows for a 9.5% reduction in the cumulative maintenance cost over 20 years, thanks to the extension of the useful life of the pavement. In practice, these savings facilitate budget planning and free up resources for other urban needs, validating the incorporation of curing as a cost-effective decision.

4. All differences between groups were statistically significant, as verified by the ANOVA, Welch ANOVA, and Kruskal–Wallis tests, with $p$-values lower than 0.05 in all cases. This quantitative validation reinforces the reliability of the results and technically supports the inclusion of curing as a relevant criterion in technical regulations, road project evaluation methodologies, and urban sustainability policies.

5. The integrative synthesis shows that curing presents the best relative performance across the three sustainability axes without generating negative trade-offs. This demonstrates that its implementation favors a balanced sustainability approach, which is strategic for cities seeking comprehensive and replicable solutions.

6. The applied methodology is replicable in other urban contexts and can be adapted to different road infrastructure scales. Furthermore, it aligns with recognized methodological frameworks such as Life Cycle Assessment (LCA), MIVES, and Envision™, strengthening its applicability in sustainable urban development plans and policies.

Overall, the results obtained support the incorporation of curing as a technical and public policy criterion in the design, maintenance, and management of urban pavements. Its systematic adoption can contribute to more resilient, efficient, and citizen-oriented cities, in line with the Sustainable Development Goals 9, 11, and 12.

**Funding:** Universidad Cooperativa de Colombia funded this research through Project INV3472.

**Institutional Review Board Statement:** Not applicable.

**Informed Consent Statement:** Not applicable.

**Data Availability Statement:** Data are available upon request to the corresponding author.

**Conflicts of Interest:** The author declares no conflicts of interest.

## Abbreviations

The following abbreviations are used in this manuscript:

| | |
|---|---|
| ANSV | Agencia Nacional de Seguridad Vial |
| COP | Colombian Peso |
| CO2 | Carbon Dioxide |
| ESALs | Equivalent Single Axle Load |
| HIPERPAV® | High PERformance concrete PAVement simulation software |
| IRI | International Roughness Index |
| kg | Kilogram |
| km | Kilometer |
| MJ | Megajoules |
| m3 | Cubic Meter |
| PCC | Portland Cement Concrete |
| PIARC | Permanent International Association of Road Congresses |
| Temp | Temperature |
| w/c | Water–Cement Ratio |

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
