# Peer review of "Curing Sustainability Assessment in Concrete Pavements: A 20-Year Simulation-Based Analysis in Urban Road Contexts"

_sustainability, doi:10.3390/su17125299_

Round 1
Reviewer 1 Report
Comments and Suggestions for Authors
In the present research, the authors try to investigate the sustainability of the concrete pavement by simulation method. Though some results are given, there are some questions.
- The authors titled the paper as “Enhancing Concrete Pavement Sustainability Through Curing: A Multidimensional 20-Year Assessment in Urban Contexts”. Based on the title, it seems that they would study the method to improve the concrete pavement sustainability. However, it is just a simulation analyses. Such a title is far from the content and results in some confusing. The authors are suggested to revise it.
- In the abstract, the authors should describe the main reason that they do such a research. The challenge or conquered problem could be illustrated using one or two sentences.
- In the introduction, the authors should illustrate the related previous researches, which could help the subsequent understanding.
- In the methodology, the authors introduce the evaluation proceeding. They claim the related data. However, there is no any supplemental document about the data. That makes the manuscript unpersuaded.
- In the introduction, the authors describe the research region. If this research is focusing, the authors are suggested to emphasize this in the title.
- In the results, the authors just provide the longitudinal cracking evolution in different pavements. It wonder whether only this factor is important? What is about others? There should be a assessment on these factors.
Author Response
Reviewer 1: We sincerely thank Reviewer 1 for the thoughtful and constructive comments. We have carefully addressed each point below and revised the manuscript accordingly.
Comments 1: The authors titled the article 'Enhancing Concrete Pavement Sustainability Through Curing: A 20-Year Multidimensional Assessment in Urban Contexts'. Judging by the title, it appears they would study a method to improve sustainability. However, this is merely a simulation analysis. The title diverges from the content and is confusing. The authors are advised to revise it.
Response 1: We thank the reviewer for this valuable observation. We acknowledge that the original title may have unintentionally suggested an experimental or field-based intervention study. To better align with the actual scope and methodology of the research, we have revised the title to:
“Curing Sustainability Assessment in Concrete Pavements: A 20-Year Simulation-Based Analysis in Urban Road Contexts”
This new title explicitly communicates that the study is based on simulation modeling (HIPERPAV®), focuses on the sustainability implications of curing, and is applied within a representative urban road environment. It accurately reflects the comparative, long-term, and multidimensional nature of the research, as well as its methodological foundations. We have ensured that this title is consistent with the abstract, introduction, and conclusions throughout the revised manuscript.
Improvement in the article: see lines 2 to 3 in yellow
Comments 2: In the abstract, the authors should describe the main reason for conducting this research. The challenge or problem being addressed can be illustrated in one or two sentences.
Response 2: Thank you for this suggestion. In response, we have revised the abstract to include a clearer statement of the research problem. The first sentence now reads:
“In urban areas with warm climates, the lack of proper curing during concrete pavement construction can significantly reduce service life, increase maintenance needs, and com-promise sustainability goals.”
This addition establishes the context and motivation of the study from the beginning.
Improvement in the article: see lines 9 to 24 in yellow
Comments 3: In the introduction, the authors should illustrate related previous research, which could support later understanding.
Response 3: We appreciate the reviewer’s suggestion regarding the inclusion of prior research. In fact, we have already incorporated a structured and up-to-date review in the Introduction (pages 1–3) that provides a solid foundation for the study. Specifically, we reference:
- Technical and durability aspects of curing, including its role in improving compressive strength, reducing shrinkage, and enhancing long-term pavement performance (e.g., Su et al., 2024; Velandia et al., 2018; Pulecio-Díaz et al., 2024 – refs. [4–8, 12]).
- Environmental implications, such as reduced COâ‚‚ emissions and construction waste due to improved durability and fewer interventions (e.g., Hou et al., 2024; Chand et al., 2015 – refs. [14, 16–18]).
- Social and economic impacts, including better road safety, increased user satisfaction, and life-cycle cost reductions associated with curing (e.g., PIARC, 2019; Santos et al., 2017; Babashamsi et al., 2022 – refs. [23–34]).
These references are strategically placed to contextualize the relevance of curing as a sustainability factor in urban pavements. They also demonstrate the scientific gap addressed by our research: the lack of a multidimensional, simulation-based assessment integrating environmental, social, and economic criteria over a long-term horizon. For this reason, no further changes were made to the Introduction, as we believe the current structure already addresses this concern.
Comments 4: in the methodology, the authors introduce the evaluation proceeding. They claim the related data. However, there is no any supplemental document about the data. That makes the manuscript unpersuaded.
Response 4: We thank the reviewer for highlighting the importance of data transparency. In response, we have prepared and uploaded a Supplementary File 1.
Comments 5: In the introduction, the authors describe the research region. If the study focuses on a specific topic, it is suggested to highlight it in the title.
Response 5: We thank the reviewer for this thoughtful observation. Although the simulation and sustainability assessment were applied to a specific location—Comuna 8 of Ibagué, Colombia—the methodology developed in this study is designed to be transferable to other urban settings with similar climatic and operational characteristics.
To address the reviewer’s suggestion, we have taken two actions:
- We have added a new paragraph in the Introduction (page 2), explicitly describing the rationale for selecting Comuna 8 as the case study and clarifying its representativeness for Latin American urban environments with medium traffic and warm climates.
- We have decided not to modify the article title by including the location. This decision was made to maintain the broader methodological relevance and international applicability of the study, in line with the scope of Sustainability. The specific context of Ibagué is now clearly stated in both the Introduction and the Abstract to ensure transparency.
We believe this approach preserves the general value of the proposed framework while acknowledging the real-world basis of the analysis.
Improvement in the article: see lines 95 to 102 in yellow
Comments 6: In the results, the authors just provide the longitudinal cracking evolution in different pavements. It wonder whether only this factor is important? What is about others? There should be a assessment on these factors.
Response 6: We thank the reviewer for this insightful comment. While longitudinal cracking was initially presented as the main structural performance indicator due to its strong sensitivity to curing practices and its relevance in estimating environmental, social, and economic impacts, we recognize the importance of evaluating other deterioration modes.
In response, we have expanded Section 3.1 to include the simulation results of three additional structural indicators: transverse cracking (%), joint faulting (cm), and ride quality (IRI, m/km). These results are presented in Figure 9, which illustrates their evolution over a 20-year horizon for both the cured and uncured scenarios.
The new analysis shows that curing provides structural benefits beyond longitudinal cracking. In particular, transverse cracking and joint faulting exhibited moderate but consistent differences favoring the cured scenario, indicating improved durability and structural integrity. In contrast, ride quality (IRI) showed minimal difference between scenarios, suggesting that surface regularity is less sensitive to curing interventions.
Importantly, among all indicators analyzed, longitudinal cracking exhibited the clearest and most sustained separation between the cured and uncured scenarios (Figure 8). This reinforces its role as the most sensitive and reliable indicator for assessing the structural impact of curing and justifies its use as the sole basis for estimating life-cycle sustainability indicators such as emissions, costs, and social impacts.
Although the additional indicators strengthen the technical discussion, their trends were less pronounced and less consistent over time. Therefore, they were included as supporting evidence, while longitudinal cracking remains the reference indicator for the sustainability evaluation.
We believe these additions enhance the comprehensiveness of the analysis and directly address the reviewer’s concern.
Improvement in the article: see lines 575 to 607 in yellow

Reviewer 2 Report
Comments and Suggestions for Authors
The manuscript presents a comprehensive study evaluating the impact of curing methods on the sustainability of concrete pavements over a 20-year period. The research focuses on environmental, social, and economic indicators, providing a multidimensional assessment of pavement performance in urban settings. The findings of the research have significant implications for urban infrastructure management, particularly in developing contexts. The demonstrated benefits of curing—such as reduced COâ‚‚ emissions, improved road safety, and lower maintenance costs—support policy recommendations for incorporating curing into pavement design and maintenance practices. The connection to SDGs further emphasizes the research's relevance to broader sustainability efforts.
As far as novelty is concerned, the research addresses a significant gap in the literature regarding the quantification of curing's effects on concrete pavement sustainability. While previous studies have acknowledged the importance of curing, this manuscript provides a rigorous, long-term analysis incorporating a range of sustainability indicators. The innovative use of both computer simulation and statistical analysis enhances the manuscript's contribution to the field, establishing a framework that can be replicated in similar contexts.
The manuscript is well-structured, with a clear flow from the introduction through the methodology, results, and discussion. The use of tables and figures aids in the presentation of complex data, making it accessible to readers. However, there are still some comments or suggestions for the author's consideration:
- Some sections could benefit from further elaboration, particularly the methodology, to ensure that readers fully understand the simulation processes and statistical tests used. For example, why were those statistical tests selected?
- The choice of sustainability indicators seems to be appropriate and relevant, covering environmental, social, and economic dimensions. Nonetheless, the manuscript could improve by providing more details on the selection criteria for the indicators and how they were measured.
- The author should clarify whether the simulation findings have been validated through sensitivity analysis. Including a sensitivity analysis would strengthen the manuscript by demonstrating how variations in input parameters affect the simulation outcomes. This would enhance the robustness of the results and provide more confidence in the conclusions drawn from the study.
Author Response
Reviewer 2: We thank Reviewer 2 for the insightful and constructive comments. Below, we provide detailed responses to each point and explain how the manuscript has been improved accordingly.
Comments 1: Some sections could benefit from further elaboration, particularly the methodology, to ensure that readers fully understand the simulation processes and statistical tests used. For example, why were those statistical tests selected?
Response 1: We appreciate this insightful observation. In response, we have expanded Sections 2.4 and 2.5.4 to clearly explain the statistical approach and the rationale for test selection.
Specifically, we clarify that each sustainability indicator was analyzed individually through a univariate comparison framework. The choice of test—one-way ANOVA, Welch’s ANOVA, or Kruskal–Wallis H—was based on the outcome of preliminary tests: Shapiro–Wilk for normality and Levene’s test for homogeneity of variances. Welch’s ANOVA was applied when variances were unequal; Kruskal–Wallis was used as a non-parametric alternative when normality was not satisfied.
This clarification now appears in the manuscript to ensure full transparency of the methodology and to support the robustness of the results.
Improvement in the article: see lines 445 to 452 in yellow
Comments 2: The choice of sustainability indicators seems to be appropriate and relevant, covering environmental, social, and economic dimensions. Nonetheless, the manuscript could improve by providing more details on the selection criteria for the indicators and how they were measured.
Response 2: We thank the reviewer for this valuable observation. In response, we have expanded Section 2.2. Selection of variables and indicators to explicitly justify the criteria used for indicator selection and describe how each indicator was measured or estimated.
We now explain that the indicators were selected based on their relevance to pavement performance, sensitivity to variations in maintenance scenarios, and alignment with international sustainability frameworks such as ISO 14040 and Envision™. Each was quantified using standardized estimation methods and expressed in consistent units per kilometer. The updated paragraph clarifies how environmental, social, and economic metrics were derived using technical literature, urban mobility studies, perception surveys, and cost estimation procedures linked to HIPERPAV® simulation outputs.
In addition, the parameter values, assumptions, and data sources for each indicator are clearly presented in Tables 2, 4, and 6 within the results section to ensure transparency and replicability.
Improvement in the article: see lines 169 to 179 in yellow
Comments 3: The author should clarify whether the simulation findings have been validated through sensitivity analysis. Including a sensitivity analysis would strengthen the manuscript by demonstrating how variations in input parameters affect the simulation outcomes. This would enhance the robustness of the results and provide more confidence in the conclusions drawn from the study.
Response 3: We appreciate this insightful recommendation. In response, we have incorporated a univariate sensitivity analysis directly into the results section (see end of Section 3.1). This analysis was performed using the actual longitudinal cracking values generated by HIPERPAV® and focused on the three most influential parameters in early-age concrete behavior: ambient temperature (24.5 °C), cement content (355 kg/m³), and water–cement (w/c) ratio (0.394). Each parameter was independently varied by ±10% to simulate typical field-level deviations under urban tropical conditions, as represented in the case study of Ibagué (Comuna 8).
The analysis results are presented in the newly added Figure 10, which compares the cracking evolution for curing and non-curing scenarios under each variation. The results confirm that while absolute values fluctuate slightly, the relative difference between scenarios remains stable throughout the 20-year pavement life. This confirms the robustness of the simulation model and supports the validity of longitudinal cracking as a reference structural indicator for sustainability assessments. These additions strengthen the reliability of the conclusions and address the reviewer’s concern regarding model sensitivity and confidence in the findings.
Improvement in the article: see lines 609 to 628 in yellow

Reviewer 3 Report
Comments and Suggestions for Authors
- Use of Bullet Points in Section 2.2.2:
- Present indicators as bullet points for clarity (e.g., accident reduction, travel time, citizen satisfaction, and job creation).
- Methodology Structure:
- Reorganize the methodology into clear subsections with headings .
- Simplify Complex Sentences:
- Shorten and simplify sentences for readability. Example: Break long descriptions into concise statements.
- Calculation Results:
- Calculations seem reasonable but consider including stepwise explanations or references for transparency.
- Figures 2 and 8:
- Add comparative labels, annotations, and clear legends.
- Correct some references format.
Author Response
Reviewer 3: …
Comments 1: Use of Bullet Points in Section 2.2.2:
Present indicators as bullet points for clarity (e.g., accident reduction, travel time, citizen satisfaction, and job creation).
Response 1: We appreciate the suggestion. Section 2.2.2 has been revised to present the social sustainability indicators in bullet point format, improving clarity and visual comprehension for the reader.
Improvement in the article: see lines 203 to 217 in yellow
Comments 2: Methodology Structure:
Reorganize the methodology into clear subsections with headings.
Response 2: Thank you for this valuable recommendation. In the revised manuscript, Section 2 has been fully reorganized into six clearly numbered subsections (2.1 to 2.6), each with specific and functional titles. Subsections now reflect the simulation process, indicator definition, data processing, and statistical analysis more transparently. This improved structure enhances clarity, facilitates reader navigation, and aligns with best practices in methodological reporting for international academic publications.
Improvement in the article: see lines 86 to 513 in yellow
Comments 3: Simplify Complex Sentences:
Shorten and simplify sentences for readability. Example: Break long descriptions into concise statements.
Response 3: We appreciate this observation. In response, we reviewed and revised several long or complex sentences throughout the manuscript—particularly in the introduction and methodology sections. The revised version adopts a more concise and direct writing style, improving clarity and readability while maintaining technical precision.
Comments 4: Calculation Results:
Calculations seem reasonable but consider including stepwise explanations or references for transparency.
Response 4: We thank the reviewer for this important observation. In the revised version, we have clarified the origin of all structural performance results presented in Section 3.1. A stepwise explanation has been added before Figure 9, indicating that each scenario (with and without curing) was independently simulated using HIPERPAV®, under the input conditions described in Section 2.3.1. Model outputs such as longitudinal and transverse cracking, joint faulting, and ride quality (IRI) were extracted directly from the software’s reporting system. This clarification enhances transparency and strengthens the traceability of the calculations.
Improvement in the article: see lines 575 to 628 in yellow
Comments 5: Figures 2 and 8:
Add comparative labels, annotations, and clear legends.
Response 5: We appreciate the reviewer’s suggestion. In the revised version of the manuscript, we have significantly improved the graphical content to enhance clarity, readability, and comparative understanding:
Figure 2 (Methodological Procedure): This diagram has been redesigned with clearer visual blocks and enlarged sections for each phase. The input data are now explicitly grouped by structural design, climatic conditions, and concrete mix parameters. The layout has been refined to improve logical flow, maintaining consistent styling throughout the article.
Figure 8 (Longitudinal Cracking): Direct comparative labels have been added to both curves (No Curing and Curing), and the Intervention Threshold (25%) is now clearly indicated with a dashed line and a descriptive annotation, allowing immediate interpretation without relying solely on the legend.
Figure 9 (Structural Performance Indicators): This set now includes three additional subfigures for Transverse Cracking, Joint Faulting, and Ride Quality (IRI). Each graph incorporates its respective technical threshold (25%, 0.30 cm, and 5 m/km) and direct labels on the base curves. Axis scales have been properly adjusted to reflect the actual data range, ensuring proportional and accurate representation.
Figure 10 (Sensitivity Analysis): Built using the same visual format, this figure integrates base curves and ±10% variations for critical input parameters. The plot follows the established aesthetic, using Palatino Linotype font and a consistent graphical structure for all simulation outputs.
Improvement in the article: see lines 138 to 154 and 554 to 628 in yellow
Comments 6: Correct some references format.
Response 6: We appreciate the reviewer’s observation regarding reference formatting. A thorough revision of all references has been conducted to ensure alignment with the citation style required by Sustainability (MDPI). This includes the correction of author names, titles, journal names, volumes, issues, page numbers, and DOI identifiers where applicable. We have also ensured consistency in format across all in-text citations and the reference list. The manuscript now meets the editorial standards of the journal.
Improvement in the article: see lines 996 to 1156 in yellow

Reviewer 4 Report
Comments and Suggestions for Authors
The paper intends to use computational simulation and statistical analysis for evaluating sustainability criteria based on the impact of curing on the service life of concrete slab pavements. The sustainability concerns seem fine in this research but the paper suffers some major drawbacks that hinder its acceptance. The details are as follows:
- The impact of curing on concrete pavement sustainability has been investigated in previous studies with field data. The simulated data of this research without field data has limited implications and scholarly value in academia.
- The paper highlights the benefits of curing but does not sufficiently explore potential trade-offs, such as the initial costs of implementing curing practices or the availability of resources in different regions.
- The write-up is poorly organised and some parts are mainly discussed on statisitical aspects of analyses, where these have been known and common in the academia. Many parts need to be re-orghanised.
Author Response
Reviewer 4: We are grateful for your thorough review and the opportunity to improve the scholarly rigor of our manuscript. Below, we provide a detailed point-by-point response to your valuable comments:
Comments 1: The impact of curing on concrete pavement sustainability has been investigated in previous studies with field data. The simulated data of this research without field data has limited implications and scholarly value in academia.
Response 1: We fully acknowledge the importance of field data in pavement sustainability assessments. However, the aim of this study was to apply a simulation-based methodology using the HIPERPAV® platform—an industry-standard tool recognized for its ability to model early-age concrete pavement behavior and long-term structural performance under specific climatic, design, and material conditions. This approach allows us to systematically isolate and analyze the impact of curing practices over a 20-year horizon, providing insights that are reproducible and generalizable across controlled urban scenarios.
To enhance the scientific robustness, we have incorporated a sensitivity analysis (see revised Section 3.1, Figure 10), evaluating how ±10% variations in critical input parameters (ambient temperature, cement content, and water-to-cement ratio) influence longitudinal cracking predictions. This addition strengthens the reliability and academic relevance of our simulation-based findings, particularly in the context of long-term sustainability analysis.
Improvement in the article: see lines 609 to 628 in yellow
Comments 2: The paper highlights the benefits of curing but does not sufficiently explore potential trade-offs, such as the initial costs of implementing curing practices or the availability of resources in different regions.
Response 2: We thank the reviewer for this valuable and constructive comment. In response, we have incorporated a new subsection entitled “4.4. Trade-offs and Contextual Constraints of Curing Implementation” (see lines 891 to 923 in yellow). This section explicitly addresses economic, logistical, and climatic trade-offs that may influence the practical adoption of curing practices in real-world infrastructure programs.
In particular, we discuss:
The short-term implementation costs related to labor, water consumption, and curing compound availability;
Regional disparities in access to curing materials and trained personnel;
The role of local climate conditions in determining the marginal value and feasibility of curing.
To support this discussion, the manuscript includes Figure 10, which presents a sensitivity analysis showing how ±10% variations in ambient temperature, cement content, and water-to-cement ratio significantly affect longitudinal cracking development. The results demonstrate that pavements without curing are more vulnerable to such variations, whereas curing stabilizes performance and mitigates early degradation. This highlights curing as a risk-reducing strategy, particularly important in contexts with limited quality control.
Furthermore, we provide Figures 11 to 13 to illustrate the cumulative sustainability impacts over a 20-year period:
Figure 11 shows modest increases in water use but substantial reductions in COâ‚‚ emissions, energy consumption, and construction waste in the curing scenario.
Figure 12 displays improvements in key social indicators, including accident reduction and travel time.
Figure 13 offers a visual synthesis of environmental, social, and economic dimensions via a radar chart, clearly confirming that curing yields superior overall performance across sustainability pillars.
These additions provide both qualitative and quantitative support to ensure a more comprehensive and context-aware analysis of curing practices. We are confident that the expanded discussion meaningfully addresses the reviewer’s concern and strengthens the applicability and policy relevance of the manuscript.
Comments 3: The write-up is poorly organised and some parts are mainly discussed on statisitical aspects of analyses, where these have been known and common in the academia. Many parts need to be re-orghanised.
Response 3: Your comment prompted a thorough revision of the manuscript’s structure and clarity. Specifically:
The Methodology section (Section 2) has been reorganized into six clearly defined subsections, using concise language and step-by-step logic to explain the simulation flow, input data, and evaluation framework.
In Section 2.2.2, we adopted bullet points for indicator definitions, improving readability.
Improvement in the article: see lines 86 to 513 in yellow
Figures 2, 8, 9, and 10 have been redesigned with comparative annotations, intervention thresholds, and clear legends to enhance data visualization and interpretation.
Improvement in the article: see lines 138 to 154 and 554 to 628 in yellow
Throughout the manuscript, we have refined the narrative flow to reduce redundancy and improve conceptual coherence, in alignment with scientific writing standards expected in Q1 journals.

Round 2
Reviewer 1 Report
Comments and Suggestions for Authors
In the present state, the authors have revised the manuscript and answered questions. Now, it is improved and could be accepted.
Reviewer 4 Report
Comments and Suggestions for Authors
The comments have been addressed accordingly. I have no further comment.